

# Dynamic upper-ocean processes enhance mesopelagic carbon export of zooplankton fecal pellets in the southern South China Sea

Ruitong Wu[1], Zhifei Liu[1], Jiaying Li[1], Baozhi Lin[1], Yulong Zhao[1], Junyuan Cao[1], and Xiaodong Zhang[1]

[1]State Key Laboratory of Marine Geology, Tongji University, Shanghai, China

5 *Correspondence*: Zhifei Liu (lzhifei@tongji.edu.cn)

**Abstract.** Zooplankton are key contributors to the marine biological pump by converting phytoplankton-derived organic carbon into fast-sinking fecal pellets. Despite the established role of upper ocean dynamics in regulating epipelagic biogeochemistry and plankton communities, their impact on mesopelagic fecal pellet carbon export remains poorly constrained. Here, we present time-series sediment trap mooring observations of fecal pellet fluxes at 500 m from August 2022 to May 10 2023 in the southern South China Sea. Zooplankton fecal pellet fluxes display distinct seasonal patterns, with average numerical and carbon fluxes of $7.39 \times 10^4$ pellets $m^{-2}$ $d^{-1}$ and 1.27 mg C $m^{-2}$ $d^{-1}$, respectively. Fecal pellets account for 10.0 to 42.6 % (average 21.6 %) of particulate organic carbon export, exceeding most oligotrophic regions. Mesopelagic fecal pellet fluxes are strongly correlated with upper-ocean dynamic processes, including winter mixing, tropical cyclones, and mesoscale eddies. Two tropical cyclones increase regional fecal pellet carbon export by more than 10 % of the annual carbon flux. One 15 spring peak contributes more than 60 % of the total flux, likely driven by the combined effects of winter mixing, cold eddy activity, and spring zooplankton blooms. Our results highlight the critical role of upper-ocean dynamics in fecal pellet carbon export in deep water layers.

## 1 Introduction

The latest Global Carbon Budget for 2024 has revealed unprecedented atmospheric $CO_2$ levels of 422.45 ppm, representing a 20 52 % increase from the preindustrial level of 278 ppm, mainly due to rising anthropogenic emissions (Friedlingstein et al., 2025). In light of this pressing crisis, the ocean plays a crucial role in mitigating climate change, acting as a significant carbon sink that absorbs $2.9 \pm 0.4$ Gt C annually through complex physical and biological mechanisms (Friedlingstein et al., 2025). The biological carbon pump (BCP), at the heart of these mechanisms, serves as an irreplaceable component for converting dissolved $CO_2$ in the surface ocean into particulate organic carbon (POC) through the photosynthesis carried out by 25 phytoplankton (Falkowski, 2012), exporting approximately 10 Pg C from the surface ocean annually and sequestering over 1300 Pg C (Boyd and Trull, 2007; Nowicki et al., 2022). In the BCP, carbon is effectively transferred from the atmosphere to the deep ocean via a network of interconnected mechanisms (Siegel et al., 2023), including gravitational carbon transport (Boyd et al., 2019; Nowicki et al., 2022), active carbon transport related to diel vertical migration (Steinberg and Landry, 2017;



Smith et al., 2025), and physical mixing processes driven by submesoscale to meridional mechanisms (Boyd et al., 2019;

Resplandy et al., 2019).

Zooplankton fecal pellets, as excrement generated by zooplankton through grazing on phytoplankton and organic matter (Steinberg and Landry, 2017), represent a crucial component of POC in the gravitational carbon flux. By transforming slow-sinking biogenic elements into densely packed fecal pellets, zooplankton significantly reduce microbial degradation and dissolution of organic matter during the sinking process, promoting regional carbon export efficiency (Turner and Ferrante,

1979; Turner, 2002). Modern methodological advances, including sediment traps and large filtering systems, have enabled precise quantification of pellet characteristics (size, shape, density) and sinking velocities across different taxa. These studies demonstrate that fecal pellets contribute substantially to the biological pump, with their proportional contribution to POC varying dramatically from < 1 % to > 100 % across diverse marine ecosystems (Turner, 2015).

Research over recent decades has substantially improved our understanding of zooplankton fecal pellet flux dynamics through

integrated approaches combining satellite observations, in situ measurements, Bio-Argo floats, and numerical modeling. These investigations reveal two primary regulatory mechanisms: (1) bottom-up control through surface primary productivity and zooplankton community structure, mediated by regional biogeochemistry and hydrological conditions, and (2) particle transformation processes during sedimentation, including microbial degradation, particle repackaging, coprophagy, and diel vertical migration of zooplankton (Turner, 2002, 2015). Distinct seasonal and episodic events, including spring phytoplankton

blooms (Dagg et al., 2003), monsoon cycles (Carroll et al., 1998; Roman et al., 2000; Ramaswamy et al., 2005), sea ice melting (Lalande et al., 2021), and El Niño events (Menschel and González, 2019), are particularly effective at enhancing mesopelagic fecal pellet export, often associated with characteristic high-flux periods.

In recent years, growing evidence has underscored the importance of upper ocean dynamics in regulating surface biogeochemistry and carbon export, especially in stratified oligotrophic systems (Dai et al., 2023). Processes including tropical

cyclones, mesoscale eddies, and mixing events can rapidly modify surface physical-chemical gradients and plankton communities, overriding bottom-up controls on fecal pellet export. Cyclonic eddies, for example, consistently increase zooplankton biomass, abundance, and active transport (Strzelecki et al., 2007; Landry et al., 2008; Labat et al., 2009; Chen et al; 2020; Belkin et al., 2022), shift regional plankton community structure (Franco et al., 2023), and drive elevated gravitational export through increased fecal pellet production (Goldthwait and Steinberg, 2008; Shatova et al., 2012; Fischer et al., 2021).

Similarly, typhoons, storms, and tropical cyclones amplify surface turbulence, triggering phytoplankton blooms and subsequent zooplankton responses that further modulate carbon export (Baek et al., 2020; Li and Tang, 2022; Chen et al., 2022, 2023; Rühl and Möller, 2024). However, current understanding remains fragmented regarding how these mechanisms affect mesopelagic (200–1000 m) fecal pellet fluxes, as most research focuses on epipelagic (0–200 m) responses and rarely quantifies fecal pellets' specific contribution to POC flux. Reported effects also exhibit variability across different ecosystems:

while some eddies increase fecal pellet export (Goldthwait and Steinberg, 2008), others show flux attenuation despite elevated





zooplankton biomass (Christiansen et al., 2018), reflecting complex dependencies on regional hydrology and zooplankton community structure.

The South China Sea (SCS), governed by the East Asian monsoon (EAM) system, represents an ideal natural laboratory for studying upper-ocean processes due to its dynamic interplay of surface mixing, mesoscale eddies, typhoons, and tropical cyclones. In this region, fecal pellets contribute substantially (20 %) to POC flux during monsoon periods, highlighting their importance in biological carbon sequestration (Li et al., 2022). While seasonal flux patterns are commonly attributed to monsoon forcing (Li et al., 2022; Wang et al., 2023; Cao et al., 2024), the specific impacts of individual physical processes on zooplankton-mediated carbon export remain poorly understood. Key questions remain unsolved: (1) How do transient processes (e.g., tropical cyclones, eddies) modify mesopelagic fecal pellet fluxes? and (2) is the EAM the dominant driver of fecal pellet carbon export across different SCS regimes? To address these gaps, we present high-resolution time-series data from sediment trap observations (August 2022 to May 2023) at mooring station TJ-S in the oligotrophic southern SCS, integrated with synchronous physical and biogeochemical data. Our study provides the first quantitative assessment of how winter-mixing, tropical cyclones and mesoscale eddies collectively regulate fecal pellet carbon export, offering new insights into BCP dynamics in monsoon-driven marginal systems.

## 2 Material and methods

### 2.1 Environmental background

The South China Sea (SCS), the largest semi-enclosed marginal sea in the western Pacific, spans $3.5 \times 10^6$ km$^2$ with an average depth of 1140 m (Wang and Li, 2009). Our study focuses on the southern SCS near the Sunda Shelf (Fig. 1a), where the East Asian monsoon system (EAM) dominates surface wind fields and ocean circulation (Shaw and Chao, 1994). Monsoon dynamics exhibit strong seasonal variability in the research area: From June to September, southwest monsoon winds establish an anticyclonic surface circulation, following trans-equatorial air from the Indian Ocean across the Indochina Peninsula (Hu et al., 2000). From November to April, strong northeast monsoon winds drive a cyclonic surface circulation while intensifying vertical mixing (Hu et al., 2000; Fang et al., 2002). In the SCS, the mixed layer is relatively deep throughout the year, resulting in stable surface temperature (SST) and salinity conditions. As a typical oligotrophic system (Wong et al., 2007; Du et al., 2017), regional primary production exhibits distinct seasonal patterns. In summer, strong thermal stratification limits nutrient upwelling, suppressing productivity, whereas in winter, enhanced wind-driven mixing entrains deep nutrients, sustaining high primary production. Zooplankton assemblages in the southern SCS are dominated by copepods (47.1 % of total species), followed by ostracods (8.4 %) and siphonophores (7.8 %), along with contributions from pteropods, euphausiids, hydrozoans, amphipods, and various larval forms (Du et al., 2014). Diatoms dominate the phytoplankton community (75 % of total abundance across all depths), followed by cyanobacteria (13.5 %) and dinoflagellates (10.8 %), collectively forming a small size assemblage (Zhu et al., 2003). Phytoplankton exhibit maximal abundance in subsurface waters (35–75 m), while



zooplankton biomass and abundance peak in the upper 200 m, both displaying a gradual decrease with depth (Zhu et al., 2003). Plankton community structure and distribution are regulated by regional geographical settings, hydrological conditions, and dynamic processes, with wind fields, water mass characteristics, and vertical mixing acting as key external drivers. In the southern SCS, these factors operate predominantly through monsoon-forced circulation patterns (Wang et al., 2015).

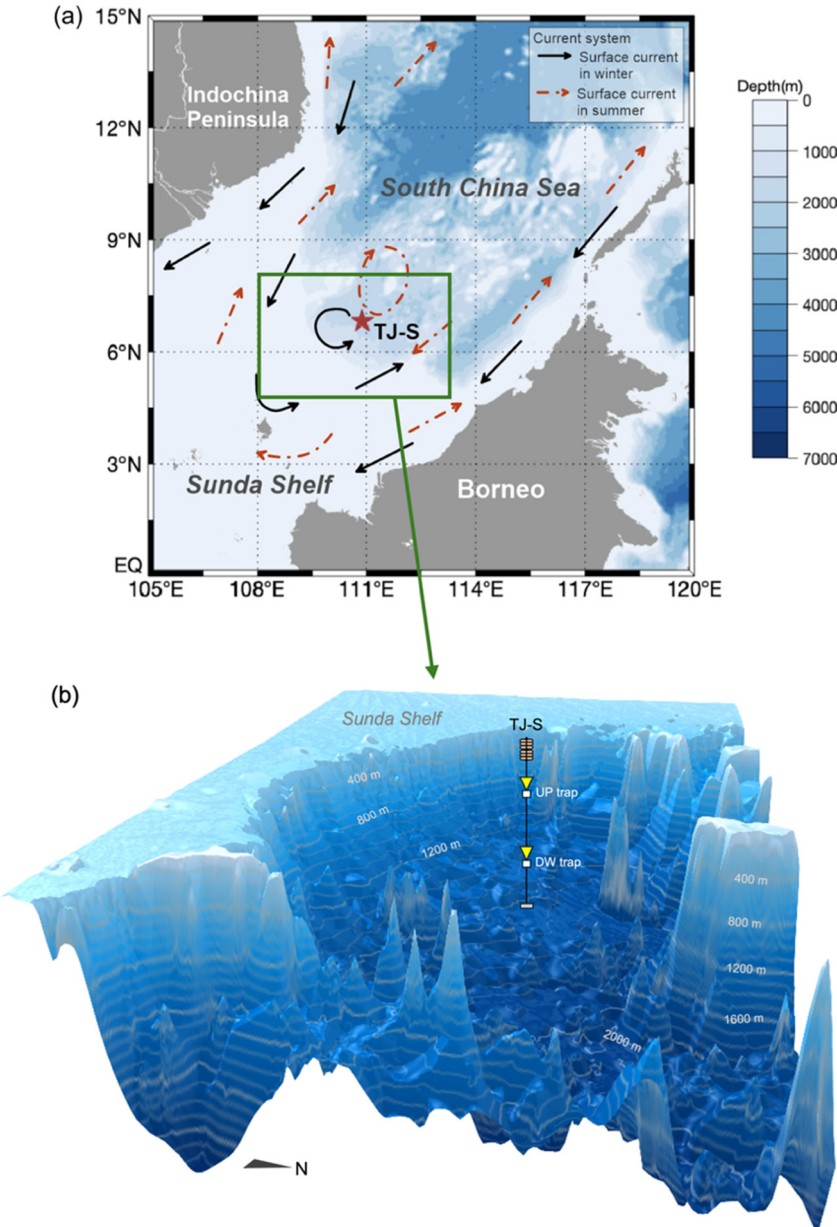

**Figure 1**. Geographic locations of the study area and the mooring system. (**a**) Surface current systems in the southern SCS, after Liu et al. (2016). Black arrows indicate surface current circulation in winter, and red dash-line arrows represent surface currents in summer. (**b**) 3D bathymetry map of the research area and vertical structures of the sediment trap mooring system TJ-S.



## 2.2 Sample processing and fecal pellet analysis

Mooring TJ-S is situated in the southern SCS, near the Sunda Shelf (6.72ºN, 110.76ºE) at a water depth of 1630 m (see Fig. 1b). Between August 2022 and May 2023, two sediment traps (UP trap and DW trap) were deployed at depths of 500 m and 1590 m, respectively, each featuring a sampling area of 0.5 m². The upper trap is equipped with 13 receiving cups, collecting samples over a 22-day interval, while the downward trap contains 22 receiving cups with a 13-day collection period. In August 2023, the sediment traps were retrieved. However, for unknown reasons, the DW trap experienced a malfunction after October 2022, resulting in only two unaffected samples from August to September. In contrast, the samples from the UP trap remained fully intact. Consequently, this study focuses primarily on the complete time-series samples obtained from the UP trap for experimental analysis and further discussion. Sample separation and pretreatment were conducted at the State Key Laboratory of Marine Geology at Tongji University via methods established by Li et al. (2022). The wet subsample (the aliquot) was sieved through a 20-μm Nitex© mesh to separate fecal pellets from finer terrigenous sediments, such as clay minerals. The retained fraction was carefully rinsed into a gridded petri dish and evenly distributed for microscopic analysis. Fecal pellets were then counted using a Zeiss Stemi 508 stereomicroscope. To standardize the measurements, fecal pellets were categorized into large (width > 100 μm) and small (width < 100 μm) pellets. Large fecal pellets were counted and photographed at 8x magnification, while small pellets were enumerated at 50x magnification, with 32 to 50 random fields captured to minimize subjective bias. In cases of densely packed samples, further subsampling (2–3 splits) was conducted prior to imaging. Morphological parameters (e.g., length and width) of fecal pellets were measured via digital software (Image J), and their biovolume was calculated using mathematical formulas based on their shapes (Li et al., 2022). The volume was then converted to carbon content using a carbon-volume conversion factor of 0.036 mg C mm⁻³, as previously reported in the southern SCS (Li et al., 2022). Fecal pellet numerical (FPN) flux (pellets m⁻² d⁻¹) and fecal pellet carbon (FPC) flux (mg C m⁻² d⁻¹) were calculated for each sample, taking into account the area of the petri dish, the photographic coverage area, the sediment trap collection area (0.5 m²), and the duration of sampling.

## 2.3 Hydrological parameter analysis

To investigate the temporal variability of zooplankton fecal pellet fluxes in sediment trap samples and the underlying mechanisms, we conducted a comprehensive analysis that incorporated several key physical and biogeochemical parameters. These parameters include surface wind fields, precipitation, sea surface temperature (SST), nutrient concentration, mixed layer depth (MLD), sea level anomaly (SLA), primary productivity (PP), and chlorophyll-a (Chla) concentration. We obtained hourly wind speed, SST, and precipitation data with a spatial resolution of 0.25° × 0.25° from the fifth-generation ECMWF reanalysis for global climate and weather (ERA5). Daily MLD data, with a spatial resolution of 0.83° × 0.83°, were sourced from the CMEMS Global Ocean Physics Analysis and Forecast. Additionally, daily SLA data were retrieved from the CMEMS Global Ocean Eddy-Resolving Reanalysis, while daily Chla and PP data were downloaded from the CMEMS Global Ocean Biogeochemistry Analysis and Forecast. For the daily mass content of zooplankton expressed in carbon (g m⁻²), we derived



data from the CMEMS Global Ocean Low and Mid Trophic Levels (LMTL) biomass content hindcast. Before proceeding with further analysis, we also calculated wind stress and determined the depth of the subsurface chlorophyll maximum (SCM). Wind stress was calculated using the formula $\tau = Cd \times \rho \times V^2$, where V represents the wind speed (m s$^{-1}$) at 10 m above the

sea surface, $\rho$ represents the air density (1.225 kg m$^{-3}$), and Cd represents the drag coefficient. The SCM depth was manually identified based on chlorophyll concentration data from CMEMS, which spans 31 depth levels from 0 m to 500 m.

## 2.4 Statistical analysis

Statistical analyses were performed using the IBM SPSS Statistics (Version 27; IBM Corp., Armonk, NY, USA). Pearson's correlation analysis was utilized to examine the consistency of environmental variables during monsoon and non-monsoon

periods. P-values were derived from two-sided t-tests to determine differences between groups with normally distributed data. Statistical significance was defined as $p < 0.05$. The Pearson correlation coefficient (R) ranges from –1 to 1, where 1 indicates a perfect positive linear correlation, –1 indicates a perfect negative linear correlation, and 0 indicates no linear correlation. Statistical graphs were generated using Grapher (Version 15; Golden Software, LLC). Marine data maps were created using MATLAB R2020a (The MathWorks, Inc., Natick, MA, USA) with the M_Map package (Version 1.4). Three-dimensional

topographic maps were produced using QGIS (Version 3.16) with the Qgis2threejs plugin (Version 2.8).

## 3 Results

### 3.1 Fecal pellet fluxes and characteristics

Three types of zooplankton fecal pellets were identified at Mooring TJ-S: spherical, cylindrical, and ellipsoidal (Fig. 2a). Most of the fecal pellets are light green, green, brown, or dark brown in color, featuring a compact structure and distinct edges.

Larger pellets tend to be darker in color, while smaller ones appear relatively transparent (Fig. 2b; detailed photographs can be found in Figs. S1–S6 in the Supplement). The geometric parameters (biovolume, length, width) and flux characteristics (FPN, FPC) of zooplankton fecal pellets at TJ-S are statistically displayed in Table 1 and Figure 3 (detailed data are available in Table S1 in the Supplement). We observe significant variations in size across fecal pellet types. Cylindrical pellets are particularly large, with an average biovolume of $6.71 \times 10^5$ $\mu$m$^3$, which is approximately 2 to 4 times greater than that of

ellipsoidal pellets, averaging $1.61 \times 10^5$ $\mu$m$^3$. Spherical pellets, while similar in size to ellipsoidal pellets, are slightly smaller, averaging $2.68 \times 10^5$ $\mu$m$^3$. There is also notable intra-type variability, especially in cylindrical pellets, which can range in length from over 1 mm and width from more than 400 $\mu$m for larger specimens to less than 50 $\mu$m in width for smaller specimens (Table 1; detailed information can be found in Text S1 and Figs. S7–S8 in the Supplement). FPN ranges from 937 pellets m$^{-2}$ d$^{-1}$ to $4.61 \times 10^5$ pellets m$^{-2}$ d$^{-1}$, with an average of $7.39 \times 10^4$ pellets m$^{-2}$ d$^{-1}$ (Fig. 3a). FPC varies from 0.03 mg C

m$^{-2}$ d$^{-1}$ to 4.62 mg C m$^{-2}$ d$^{-1}$, averaging 0.91 mg C m$^{-2}$ d$^{-1}$ (Fig. 3d).



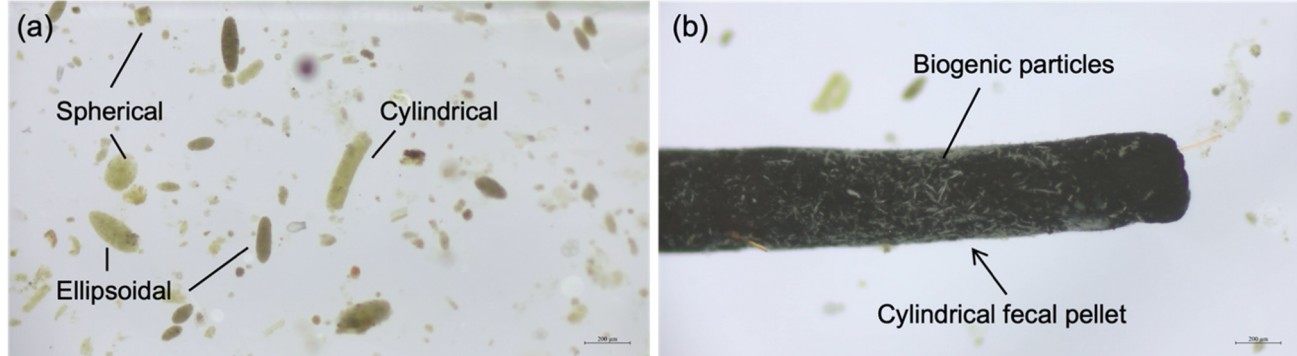

**Figure 2**. Optical micrograph of fecal pellets from typical samples at TJ-S in the southern SCS. (**a**) Three types of fecal pellets (spherical, cylindrical, and ellipsoidal). (**b**) A carbon-rich cylindrical fecal pellet with biogenic particles attached to its surface.

Fecal pellet fluxes at TJ-S exhibit pronounced seasonal variations, with both FPN and FPC showing similar time-series patterns. During August and September, both fluxes were relatively low, reaching minimum values of only 937 pellets $m^{-2}$ $d^{-1}$ and 0.03 mg C $m^{-2}$ $d^{-1}$ occurring in mid-September, respectively. From October to February, both fluxes steadily increased, peaking at the end of February. During this peak, FPN flux reached $1.78 \times 10^5$ pellets $m^{-2}$ $d^{-1}$, while FPC reached 2.14 mg C $m^{-2}$ $d^{-1}$. Additionally, there was a peak in late November when FPN surged to $8.57 \times 10^4$ pellets $m^{-2}$ $d^{-1}$, and FPC reached 1.61 mg C

$m^{-2}$ $d^{-1}$, which was 4–5 times higher than adjacent samples. In early March, fluxes declined, with FPN dropping to $2.98 \times 10^4$ pellets $m^{-2}$ $d^{-1}$ and FPC decreasing to 0.58 mg C $m^{-2}$ $d^{-1}$. However, there was a sharp increase in both FPN and FPC fluxes in mid-March, when FPN surged to $4.61 \times 10^5$ pellets $m^{-2}$ $d^{-1}$ and FPC reached 4.62 mg C $m^{-2}$ $d^{-1}$. This peak in March was 2 to 3 times greater than the peaks observed in late February and late November and was 10 to 17 times higher than the adjacent samples, marking the annual maximum. From April to May, both FPN and FPC fluxes decreased to an average of $2.61 \times 10^4$

pellets $m^{-2}$ $d^{-1}$ and 0.52 mg C $m^{-2}$ $d^{-1}$, respectively, with a slightly increasing trend noted in May.

The three types of fecal pellets exhibited distinct contributions to both FPN and FPC flux. Ellipsoidal pellets were the most prevalent ($3.38 \times 10^4$ pellets $m^{-2}$ $d^{-1}$, 48 % of total FPN), followed by spherical ($2.62 \times 10^4$ pellets $m^{-2}$ $d^{-1}$, 38%) and cylindrical pellets ($1.39 \times 10^4$ pellets $m^{-2}$ $d^{-1}$, 15 %), with the first two types collectively representing over 90 % of total FPN. In contrast, carbon flux contributions revealed a different pattern. Despite numerical scarcity, cylindrical pellets contributed

disproportionately to FPC (38 % of the total) due to their substantially larger biovolume. Ellipsoidal pellets, though numerically dominant, accounted for 34 % of FPC, while spherical pellets represented only 28 %. The size-driven disparity was particularly evident during low-flux periods. In late September, when the total pellet flux reached its minimum, large cylindrical pellets contributed over 60 % of FPC flux despite their scarcity.




**Table 1.** Geometric parameters and fecal pellet fluxes at TJ-S (500 m) in the southern SCS.

| Fecal pellet type | Number measured | Length (μm) | Width (μm) | Biovolume (×10⁵ μm³) | FPN flux (×10⁴ m⁻² d⁻¹) | FPN percentage (%) | FPC flux (mg C m⁻² d⁻¹) | FPC percentage (%) |
|---|---|---|---|---|---|---|---|---|
| Ellipsoidal | 1321 | 56–776 | 30–422 | 0.03–63.03 | 0.04–21.08 | 37–56 | 0.01–1.53 | 10–47 |
| | | **197** | **85** | **1.61** | **3.38** | **48** | **0.32** | **34** |
| Cylindrical | 1167 | 59–3487 | 20–722 | 0.02–547.92 | 0.01–9.87 | 7–21 | 0.01–2.19 | 21–59 |
| | | **322** | **84** | **6.71** | **1.39** | **15** | **0.38** | **38** |
| Spherical | 1352 | | 18–931 | 0.01–422.90 | 0.04–15.15 | 31–46 | 0.01–0.90 | 19–42 |
| | | | **104** | **2.68** | **2.62** | **38** | **0.22** | **28** |
| Total | 3840 | 18–3487 | 18–931 | 0.01–547.92 | 0.09–46.10 | 100 | 0.03–4.62 | 100 |
| | | **202** | **92** | **3.54** | **7.39** | | **0.91** | |

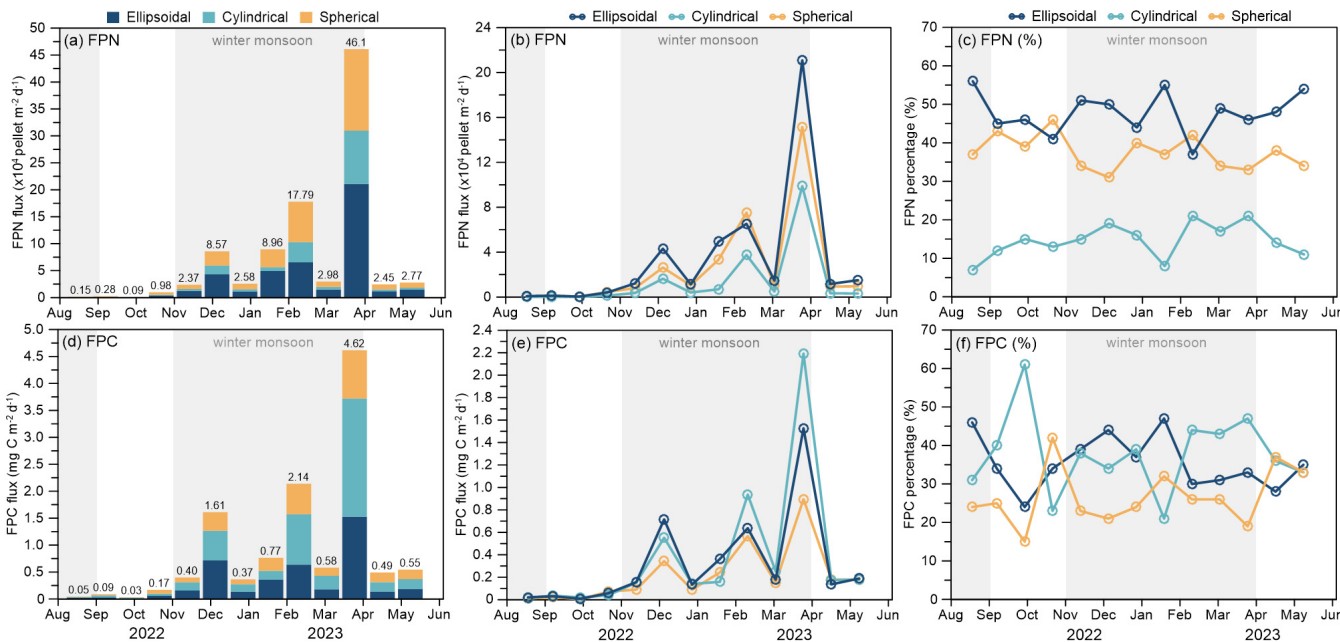

**Figure 3.** Time-series variation of FPN and FPC at sediment trap Mooring TJ-S (500 m) in the southern SCS. (**a, b**) FPN flux; (**d, e**) FPC flux; (**c, f**) FPN and FPC percentages of three types of fecal pellets. Gray bars indicate monsoon periods from August to September (summer monsoon) and November to April (winter monsoon).

## 3.3 POC flux and FPC/POC ratio

During the sampling period, POC fluxes varied significantly, ranging from 0.08 mg C m⁻² d⁻¹ to 15.97 mg C m⁻² d⁻¹, with an average of 4.56 mg C m⁻² d⁻¹, displaying seasonal variations that closely mirrored the patterns of FPN and FPC (Fig. 4). The seasonal progression showed minimum values in late summer (0.08 mg C m⁻² d⁻¹), followed by a gradual increase from October




to February, reaching a peak at 9.50 mg C m$^{-2}$ d$^{-1}$ in February. A peak of 11.38 mg C m$^{-2}$ d$^{-1}$ was observed in late November, which was approximately 4 to 8 times higher than adjacent samples, surpassing the February peak. After a temporary decline to 3.83 mg C m$^{-2}$ d$^{-1}$ in early March, fluxes surged to an annual maximum of 15.97 mg C m$^{-2}$ d$^{-1}$ by the end of the month before stabilizing at lower levels (average 4.13 mg C m$^{-2}$ d$^{-1}$) from April to May. The contribution of FPC to POC at 500 m fluctuated

between 10.0 % and 42.6 % (with an average of 21.6 %), displaying inverse seasonal variations (Fig. 4a, e). The FPC/POC ratio remained elevated from August to October during summer, averaging around 31 %, and declined during the winter monsoon, with peak values occurring in November (30.2 %), February (22.5 %), and late March (28.9 %).

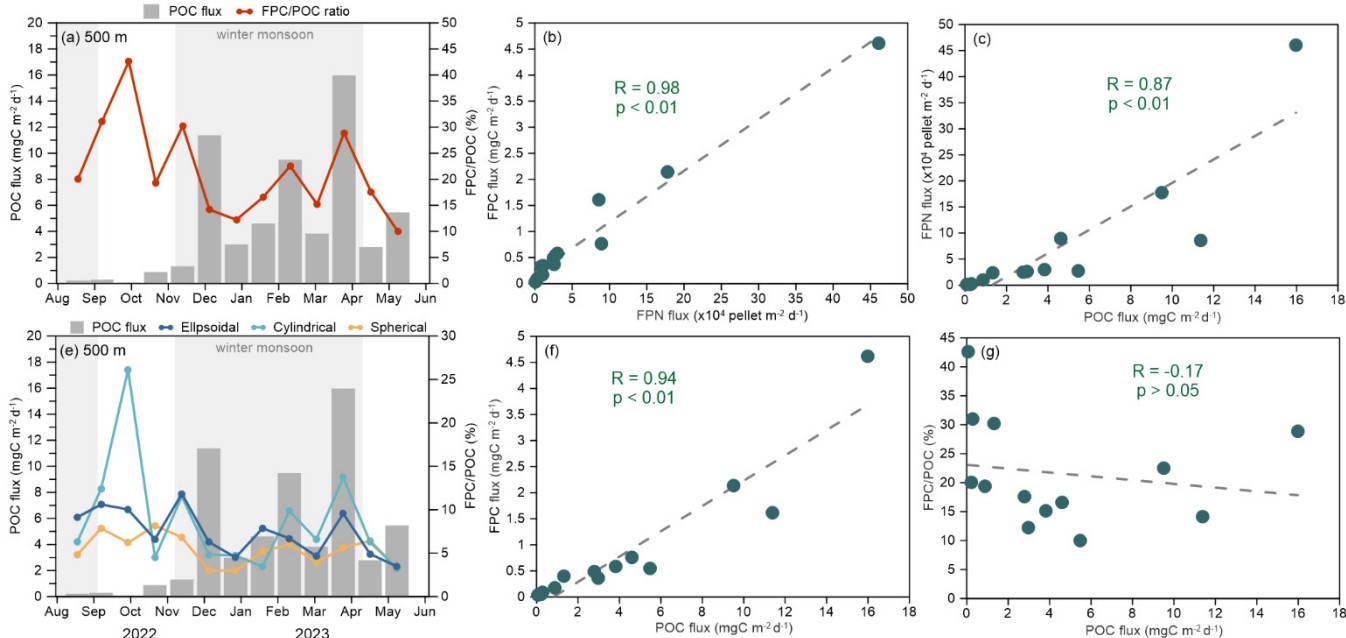

**Figure 4.** POC flux and FPC/POC ratio at Mooring TJ-S in the southern SCS (**a**) Total POC flux and FPC/POC ratio; (**b, c, f, g**)
correlations between FPC flux, FPN flux, POC flux, and FPC/POC ratio. Dashed lines indicate a linear correlation with a coefficient of R;
(**e**) total POC flux and different contributions for three types of fecal pellets. Grey bars indicate monsoon periods from August to
September (summer monsoon) and November to April (winter monsoon).

## 3.4 Upper-ocean processes during the sampling period

Physical and biogeochemical parameters are summarized in Figure 5. From August 2022 to May 2023, sea surface wind fields
at TJ-S have exhibited distinct seasonal characteristics (Fig. 5a, b). During August and September, the study area is dominated by south-westerly winds with relatively low wind speeds, averaging 3.76 m s$^{-1}$. Starting in October, wind direction gradually shifted to the northeast, and wind speed progressively increased, reaching a range of 6.91–7.95 m s$^{-1}$ from December to March. Maximum wind speed occurred at the end of January, peaking at 13.8 m s$^{-1}$. From late March to April, wind speed decreased to approximately 3 m s$^{-1}$, and wind direction gradually turned southward. Thus, the winter monsoon from November to April
is determined. Wind stress varied from 0.0001–0.4273 N m$^{-2}$, with three pronounced peaks (wind stress peaks, WSPs)



occurring during the EAM period (Fig. 5c). The first peak occurred during late December (WSP1, 0.27 N m$^{-2}$), followed by an annual maximum (WSP2, 0.43 N m$^{-2}$) in late January and a secondary peak (WSP3, 0.31 N m$^{-2}$) in early March. Two notable tropical cyclone events, TC1 and TC2, were identified based on wind direction and horizontal wind field observations (Fig. 5d). TC1 passed the mooring station during 18–22 November 2022, while TC2 was observed near Mooring TJ-S between 05–
09 January 2023 (Fig. 5d, see Fig. 8 in discussion). SST ranged from 26.1°C to 30.7°C, also displaying similar seasonal variations (Fig. 5e). From August to October, SST remained relatively stable, averaging 29.4°C. Starting in November, SST exhibited a noticeable decline, fluctuating downward from December to March, reaching the lowest (26.1°C) in mid-March. From late March onward, SST rapidly rebounded, with the average temperature in April returning to 29.5°C and peaking at 30.7°C in mid-May. The MLD varied between 10.14 m and 50.35 m, averaging 18.48 m, remaining generally consistent with
variations of surface wind field and WSPs, with several deepening events occurring during the winter monsoon (Fig. 5f). Based on reanalysis data at various depths, average primary productivity (PP) in the surface layer (0–100 m) is calculated (Fig. 5h). From August to October, surface PP values remained low, averaging 6.72 mg m$^{-3}$ d$^{-1}$. PP gradually increased in November and began to decline again in March of the following year, showing consistency with the EAM. Four significant peaks were observed, including peaks occurring in late November (22.83 mg m$^{-3}$ d$^{-1}$), early January (21.92 mg m$^{-3}$ d$^{-1}$), early February
(23.21 mg m$^{-3}$ d$^{-1}$), and late February (19.73 mg m$^{-3}$ d$^{-1}$), well correlated with tropical cyclones and MLD deepening events. Sea level anomaly at the station exhibited strong eddy activities during the sampling period (Fig. 5i). SLA values fell below zero during late March, representing a cold eddy event (see Fig.9 in discussion).

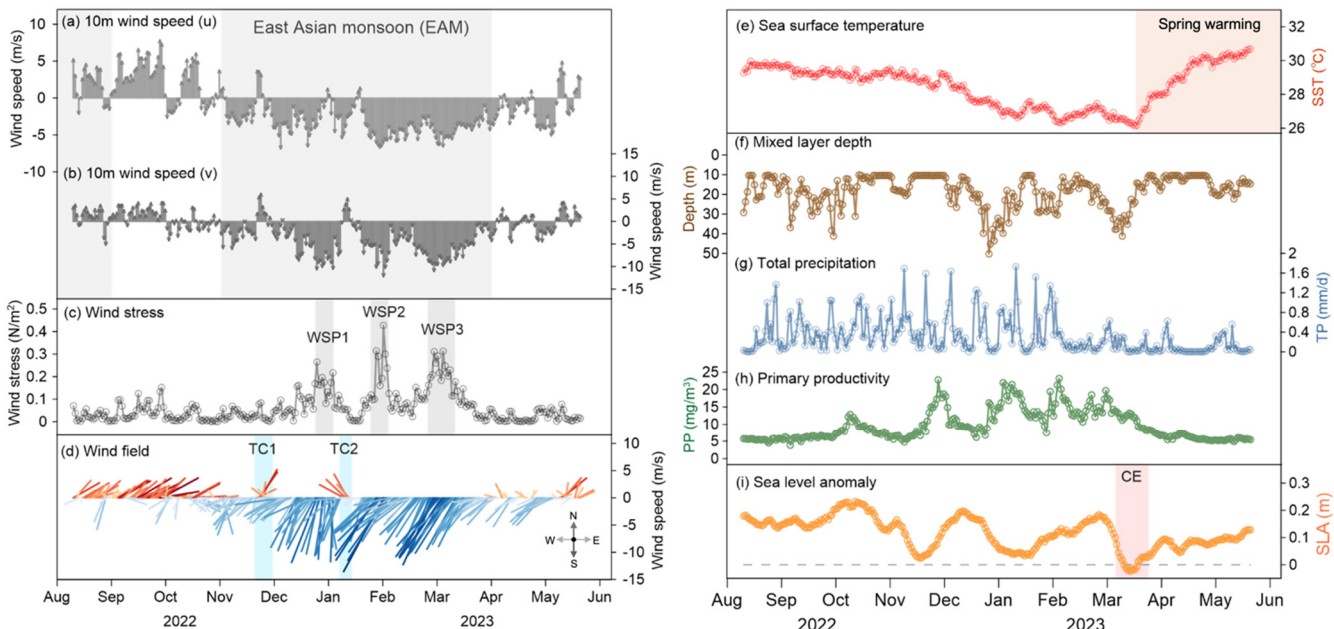

**Figure 5.** Hydrological parameters at the sediment trap Mooring TJ-S in the southern SCS. (**a**) 10 m wind speed (u); (**b**) 10 m wind speed
(v); (**c**) wind stress; (**d**) wind speed; (**e**) sea surface temperature (SST); (**f**) mixed layer depth (MLD); (**g**) total precipitation; (**h**) primary productivity; (**i**) sea level anomaly.





## 4 Discussion

In oligotrophic systems like the SCS, the dynamics of the upper ocean can override "bottom-up" controls on the export of fecal pellets by rapidly changing physical-chemical gradients as well as zooplankton behaviors. Generally, higher levels of

zooplankton fecal pellet flux are associated with phytoplankton blooms, which is supported by increases in nutrient concentration, primary production, and chlorophyll-a concentration during periods of elevated fluxes (Huffard et al., 2020). The concentration of nutrients and chlorophyll in surface waters is primarily regulated by vertical dynamics in the upper water column. Processes such as upwelling and vertical mixing are widely recognized for enhancing primary productivity by bringing nutrient-rich deep waters to the surface (McGillicuddy et al., 1999; van Ruth et al., 2010). Several hypotheses have been

proposed to explain the initiation of phytoplankton blooms. These include the critical depth hypothesis (Sverdrup, 1953; Bishop et al., 1986; Siegel et al., 2002), the critical turbulence hypothesis (Huisman et al., 1999; Waniek, 2003), and the dilution recoupling hypothesis (Behrenfeld et al., 2013). Phytoplankton blooms are often triggered when the seasonal surface mixed layer is established above critical depth or when turbulence in the surface mixed layer creates favorable light conditions.

In the southern SCS, the factors influencing the flux of zooplankton fecal pellets are quite complex. These fluxes are primarily

determined by surface primary productivity and the structure of the zooplankton community, both of which are affected by regional biogeochemical elements and hydrological conditions. The processes that regulate the settling of pellets, such as microbial remineralization and degradation, are crucial but challenging to measure due to the lack of comparisons with downward samples. Additionally, the potential role of lateral transport should not be overlooked. Strong lateral transport has been documented in the SCS, while related studies have demonstrated that most sinking organic carbon originates from surface

primary production with minimal lateral influence (Zhang et al., 2019, 2022). Here, we evaluate the impacts of several key dynamic drivers on fecal pellet carbon export in the upper (500 m) water volume at Mooring TJ-S. These drivers include surface mixing associated with the winter monsoon (Section 4.1), typhoons and tropical cyclones (4.2), and mesoscale eddies (4.3). We also discuss the potential impacts of other mechanisms, such as lateral transport and spring zooplankton blooms.

### 4.1 Contribution of winter-mixing related to the EAM system

Previous studies have demonstrated that FPN and FPC in the SCS exhibit pronounced seasonal variability and are primarily modulated by regional monsoon dynamics, with peak fluxes typically observed in winter and minimal fluxes in summer, primarily governed by the EAM system (Li et al., 2022, 2025; Wang et al., 2023; Cao et al., 2024). In their studies, this seasonal pattern correlates well with monsoon-driven variations in both wind speed and mixed layer depth. During summer, the SCS experiences persistently high SST, weak winds, strong stratification, and a deeper mixed layer, restricting the upward supply

of nutrients from deeper water. These conditions suppress primary productivity and chlorophyll concentrations, resulting in low zooplankton biomass and reduced fecal pellet flux. In contrast, strong northeasterly winds during the winter monsoon combined with large-scale cooling can intensify surface vertical mixing and promote the upwelling of deeper nutrients, which can further trigger phytoplankton blooms, leading to higher fecal pellet production and carbon export.



In our study, the extent of surface turbulence mixing and the role of the EAM is demonstrated by the combination of wind
stress and mixed layer depth. At TJ-S, the correlation analysis of hydrological parameters strongly supports the monsoon-
driven hypothesis introduced in the previous studies (Fig. 6). MLD in the southern SCS strongly depended on wind stress,
with significant positive linear relationships during both winter monsoon (R = 0.67, p < 0.01, Fig. 6b) and non-monsoon
seasons (R = 0.54, p < 0.01, Fig. 6a). Both surface nitrate and Chla concentration exhibited positive linear relationships with
MLD during the EAM (Fig. 6d, f) but displayed a negative trend during non-monsoon seasons (Fig. 6c, e). Zooplankton
biomass simulations revealed a significant positive linear relationship with surface nitrate during winter monsoon (R = 0.29,
p < 0.01, Fig. 6h), whereas a significant negative relationship was observed during non-monsoon seasons (R = –0.32, p < 0.01,
Fig. 6h).

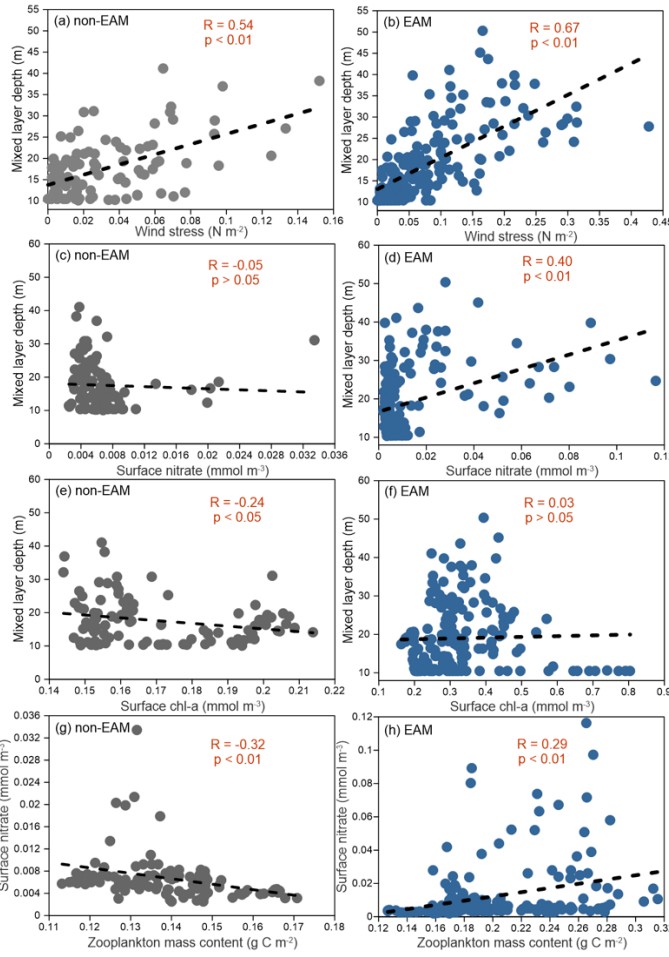

**Figure 6.** Correlation between hydrological parameters during monsoon and non-monsoon periods. (**a, b**) Wind speed and mixed layer
depth during non-EAM and EAM; (**c, d**) surface nitrate and mixed layer depth during non-EAM and EAM; (**e, f**) surface Chla and mixed
layer depth during non-EAM and EAM; (**g-h**) zooplankton mass content simulation and surface nitrate during non-EAM and EAM.
Dashed lines indicate a linear correlation with a coefficient of R.



At TJ-S, FPN and FPC exhibited general monsoon-regulated seasonal patterns, with relatively low values in summer and elevated fluxes in winter (Fig. 7g). Pronounced peaks (PKs) of pellet fluxes occurred in late November (PK1), mid-February (PK2), and late March (PK3), respectively. Notably, three pronounced wind stress peaks (WSPs) were observed during the same period (Fig. 7b), each peak well correlated with a deepening MLD, a shallowing SCM, and an increased nitrate concentration. According to the monsoon-driven hypothesis, the intensified vertical mixing during winter monsoon can facilitate effective nutrient replenishment from deeper waters, supporting elevated surface primary productivity and chlorophyll concentrations indicative of phytoplankton blooms, and under high-food-availability conditions, zooplankton biomass is able to increase significantly, resulting in rich fecal pellet production (Fig. 7f). We believe that if mooring TJ-S follows the monsoon-driven hypothesis, FPC and FPN should exhibit a good correlation with WSP events, with the annual maximum occurring during periods of the highest wind stress and the highest zooplankton biomass (WSP2). The response time depends on the complexity of the surface ecosystem, zooplankton community structure, and food chain effects in the research area. The plankton community in the southern SCS features a small class community, thus often resulting in a longer food chain (Bao et al., 2023). Thus, a time lag of several days to several weeks is expected.

Considering the possible time lag and the reception time (22 d) of sediment trap samples, we believe that PK2 and PK3 are temporally correlated with the WSPs. PK2 has a 1–23 d time lag after WSP2, whereas PK3 has a 10–32 d time lag after WSP3, both of which fall within the range of our previous assumption. PK1 is closely correlated with the SCM elevation and the increase of zooplankton biomass but has no relationship with WSP1, thus is not the result of monsoon mixing. Contrary to our expectations, the annual maximum flux (PK3) does not coincide with the peak winter monsoon period (WSP2) but is rather closely related to the second winter-mixing peak (WSP3). Wind stress, zooplankton biomass, and surface nitrate concentration in WSP3 are lower than WSP2. Thus, PK3 should theoretically result in lower values with diminished winter monsoon influence. However, PK3 significantly exceeds PK2 during the high-wind period from December to February. PK3 alone accounts for 60 % of the annual FPC export, representing a tenfold increase compared to adjacent samples. Such spring flux maxima are not observed in the northern and western SCS. At Mooring TJ-T in the southern SCS, a modest spring increase in FPC was detected. However, its magnitude remained substantially lower than the winter monsoon-driven peak, confirming the winter monsoon's dominant role in these regions (Li et al., 2022). In our mooring station, though the annual maximum does not coincide with the strongest wind stress, FPC and FPN still dominate during winter monsoon periods, accounting for over 90 % of the annual flux from November to April. Our observations support the previous theory that EAM plays a dominant role in the zooplankton fecal pellet carbon export in the research area. However, EAM alone cannot adequately explain the exceptionally high spring fluxes in March (PK3) and the first peak in October (PK1), indicating the necessity to investigate alternative explanations.







**Figure 7.** Hydrological parameters at Mooring TJ-S. (**a**) 10 m wind speed; (**b**) wind stress; (**c**) MLD; (**d**) SCM; (**e**) surface (0 m) nitrate concentration; (**f**) zooplankton mass content expressed in carbon; (**g**) FPN and FPC flux. Light grey bars represent seasonal monsoons, as shown in Fig. 3. Dark grey bars indicate the presence of wind stress peaks (WSPs).



## 4.2 Impacts of typhoons and tropical cyclones

Previous studies have demonstrated the crucial role of typhoons and tropical cyclones in inducing phytoplankton blooms and promoting deep-sea carbon export. These processes are widely acknowledged to significantly influence local biological pump processes during their passages, with surface wind stress injecting substantial energy into the upper ocean, inducing pronounced turbulent mixing, entrainment and Ekman pumping (Li and Tang, 2022), which can further transport deep water with rich nutrients to the euphotic zone, increasing primary production and organic carbon export (Zhao et al., 2008; Subrahmanyam et al., 2002; Lu et al., 2020). Over the past 23 years, approximately 83 % (92 %) of typhoons (tropical cyclones) in the SCS have been reported to effectively promote phytoplankton blooms and chlorophyll concentrations, leading to an average increase of 0.13 (0.07) mg m$^{-3}$ (Li and Tang, 2022). The magnitude of cyclonic-induced chlorophyll blooms primarily depends on their characteristics, including intensity (represented by maximum wind speed, WS) and transition speed (TS) (Sun et al., 2010; Zhao et al., 2017). WS determines the spatial extent of their impact, while TS governs the temporal duration of vertical mixing, affecting both the maximum depth and spatial coverage of phytoplankton blooms. On the continental shelf, cyclones with both high wind speeds and fast movement (WS > 25 m s$^{-1}$, TS > 5 m s$^{-1}$) as well as those with low speeds and low movement (WS < 25 m s$^{-1}$, TS < 5 m s$^{-1}$) are reported to generate greater phytoplankton blooms. In open areas, the highest chlorophyll concentrations are typically generated with cyclones with high speeds and slow movement (WS > 25 m s$^{-1}$, TS < 5 m s$^{-1}$) (Li and Tang, 2022).

To evaluate the contribution of tropical cyclones at our station, a detailed analysis of the surface wind field at Mooring TJ-S during the study period has been conducted (Fig. 8). On the basis of wind direction and horizontal wind field observations, two notable tropical cyclone events, TC1 and TC2, were identified (Fig. 8a–d). WS of the cyclones is determined by satellite data analysis, and TS is approximated with the moving distance estimated via the Haversine function. TC1 passed the mooring station from 18–22 November 2022, with a WS of approximately 20 m s$^{-1}$ and a TS of 5 m s$^{-1}$, which belongs to a slow-moving TC with slow wind speed. Shortly after the passage of TC1, significant increases in chlorophyll concentration and a shoaling of the CMD were observed (Fig. 8i). Chlorophyll concentration in the upper 50 m surged to 5.75 mmol m$^{-3}$, 2 to 3 times higher than before and after the tropical cyclone, and this peak persisted for nearly one week after its passage. Pronounced peaks in both FPN and FPC fluxes were observed in corresponding sediment trap samples (UP06, 22 November to 14 December), reaching 4–5 times the values of adjacent samples and accounting for 10 to 12 % of the annual flux (Fig. 8l). Opal fluxes in this sample also reached the highest record for the entire year (Fig. 8k). Analysis of other hydrological parameters suggest that EAM was relatively weak at this time, indicating that these observed peaks are primarily induced by the passage of TC1. TC2 was observed near Mooring TJ-S between 05–09 January 2023 (Fig. 8e–h). In contrast to TC1, TC2 did not directly pass the mooring station but remained active in the southwest region, leaving TJ-S on its periphery. An increased chlorophyll concentration and shoaling CMD were also observed (Fig. 8j) during its passage, and the corresponding sediment trap samples also received higher fluxes. Our study revealed that the passage of typhoons or tropical cyclones can directly lead to phytoplankton blooms and can effectively increase fecal pellet carbon transport. FPC induced by a single tropical cyclone





350  (TC1) can account for 10.5 % of the annual total. However, no tropical cyclones or typhoons were observed from February to March, so we still need to look for other possibilities for the spring peak.

**Figure 8.** The possible impact of tropical cyclones at TJ-S. (a–d) Surface wind field from 18–20 November 2022, representing the passage of TC1. (e, f) Surface wind field from 05–09 January 2023, representing the possible impact of TC2. (i, j) Chlorophyll concentration
355  during the passage of TC1 and TC2. (k) Opal flux. (l) Fecal pellet flux. The light grey bars represent the seasonal monsoon, as shown in Fig. 3. The dark grey bars indicate the WSPs shown in Fig.7. The blue bars represent the two tropical cyclone events.





## 4.3 Potential impact of mesoscale eddy activities

At Mooring TJ-S, strong eddy activities were observed around mid-March (Fig. 9). A cold eddy (cyclonic eddy, CE) formed near the mooring station around 06 March 2023 (Fig. 9a) and subsequently propagated north-eastward before dissipating in the northwestern waters around 22 March (Fig. 9h). A corresponding warm eddy (anticyclonic eddy, ACE) developed southeast of the station along the Borneo coast following the formation of the CE, gradually moving southwest.

This eddy event at our station coincides temporally with the spring peak in zooplankton fecal pellet flux recorded in sediment trap samples (UP11, reception time: 12 March–03 April, Fig. 9j). Recent studies suggest that dynamic mechanisms of mesoscale eddies can carry large volumes of high-kinetic-energy and thermally anomalous water masses during their movement. The horizontal advection and vertical pumping processes associated with these eddies can significantly influence regional hydrographic structures, current distributions, nutrient concentrations, and primary productivity (Chelton et al., 2011; Parker, 1971; Richardson, 1980). CEs are widely recognized to induce dome-like uplift of isopycnal layer, enhance vertical mixing, increase water column instability, and promote upward nutrient transport from deeper layers, which can further effectively replenish surface nutrients, triggering phytoplankton blooms and increase primary productivity (Xiu and Chai, 2011; Falkowski et al., 1991; McGillicuddy et al., 1999; Benitez-Nelson et al., 2007; Siegel et al., 1999; Garçon et al., 2001; Jadhav and Smitha, 2024). In contrast, ACEs are generally believed to deepen isopycnals and cannot stimulate primary productivity (Xiu and Chai, 2011; Gaube et al., 2013). Recent studies have revealed that mesoscale eddies modulate sea surface chlorophyll concentrations and phytoplankton distributions through complex interacting mechanisms (Chelton et al., 2011; McGillicuddy et al., 2007; Gaube et al., 2014; Siegel et al., 2011), including advective transport via eddy rotation (Chelton et al., 2011), entrainment of surrounding water masses and particulates at eddy peripheries (Flierl and Davis, 1993; Early et al., 2011), vertical circulation driven by eddy instability and wind forcing (Martin and Richards, 2001), and Ekman transport (Siegel et al., 2008; Gaube et al., 2014). In specific situations, elevated Chla concentrations are also observed in ACEs. Notably, frontal zones at eddy margins with high current velocity are reported to generate sub-mesoscale upwelling through intense shear forces, thus facilitating the rise of nutrient-rich deep water (Siegel et al., 2011). When CEs rotate around ACEs, kinetic energy effects can vertically induce nitrite-enriched water into the euphoric zone, increasing phytoplankton biomass.

From 14 March, TJ-S remains persistently located at the confluence of the frontal zones between these two counter-rotating eddies. Thus, we assume that these eddy activities significantly enhance regional vertical mixing with their high-velocity shear-generating upwelling that can alter subsurface nutrient distributions, which results in favorable conditions for plankton blooms. Analysis of the eddies' trajectories suggests that they can likely facilitate water mass exchange between the Mekong River plume (north) and Borneo coastal waters (south), potentially entraining nutrients, particulate matter, and plankton communities. A slight elevation of the SCM is observed from 09–17 March (Fig. 9i). Zooplankton mass content, opal flux, and POC flux also increase during the eddy activity. MLD during WSP3 is deeper than the winter peak (WSP2), suggesting the possible interactions between monsoon mixing and mesoscale eddies. In the SCS, the combined effects of cyclonic eddy activity and monsoon-induced vertical mixing are reported to significantly increase the efficiency of regional biological pumps, as





evidenced by elevated POC and opal fluxes during eddy activities (Li et al., 2017). We assume that this spring eddy activity
may have laterally advected fecal pellets from surrounding waters, contributing to carbon transport through physical transport
rather than contributing to in situ fecal pellet production. Despite peak fecal pellet fluxes and zooplankton biomass, neither
Chla concentration nor primary production showed elevated values during this period. This pattern likely reflects strong top-
down control, where intense zooplankton grazing pressure suppressed the standing stocks of phytoplankton, a phenomenon
well documented in high-nutrient, low-chlorophyll regions (Gervais et al., 2002; Schultes et al., 2006; Henjes et al., 2007).



**Figure 9.** The possible impact of mesoscale eddy activities at Mooring TJ-S. (**a–h**) Sea surface anomaly and current field from 06-22
March 2023. The blue star represents the location of TJ-S. (**i**) Chla concentration during eddy activity. (**j**) FPN and FPC fluxes. The light
grey, dark grey, and blue bars represent winter monsoon, WSPs, and TCs, respectively, as shown in Fig. 5. Red bars indicate possible eddy
activities.





## 5 Concluding remarks

In this study, we address comprehensive studies on zooplankton fecal pellet characteristics, seasonal flux variations, and controlling mechanisms by using sediment trap samples and open-source hydrological data. In the southern SCS, both FPN and FPC fluxes display distinct seasonal patterns, with minimum values occurring in late September and increasing values
from October to February. The EAM system plays a dominant role in regional fecal pellet production, evidenced by seasonal variations and corresponding changes in wind stress, MLD, SCM, and fecal pellet flux. In contrast to traditional paradigms, rather than the winter peak, the spring annual maximum makes the greatest contribution, indicating the possible contribution of other physical processes. At Mooring TJ-S, high zooplankton fecal pellet fluxes result from the combined mechanisms of winter mixing, tropical cyclones, eddy activities, and spring zooplankton blooms. Cyclone-induced fluxes account for over
10 % of the annual total. The spring annual maximum contributes more than 60 % of the total flux, likely resulting from the combined effects of the EAM, eddy activity, and temperature-zooplankton interaction. In the SCS, zooplankton fecal pellets make the most contribution to POC export in the southern region. The FPC/POC ratio in this study ranges from 10.0 % to 42.6 %, reaching an average of 21.6 % and exceeding most oligotrophic regions, demonstrating the critical role of zooplankton fecal pellets in the unique carbon export process in the southern SCS.

**Data availability**

Data of zooplankton fecal pellets and particulate organic carbon generated by this study can be found in the Supplement.

**Supplement**

The supplement will be published alongside this article.

**Author contributions**

ZL designed the study and obtained the funding. RW carried out the measurements and wrote the original draft with help of ZL, JL, BL, and JC. ZL, JL, BL, YZ, JC, XZ, and RW participated in mooring deployment/recovery cruises.

**Competing interests**

The contact author has declared that none of the authors has any competing interests.



**Disclaimer**

Publisher's note: Copernicus Publications remains neutral with regard to jurisdictional claims made in the text, published maps, institutional affiliations, or any other geographical representation in this paper. While Copernicus Publications makes every effort to include appropriate place names, the final responsibility lies with the authors.

**Acknowledgements**

We would like to thank Hongzhe Song and Wenzhuo Wang for their assistance during the laboratory analysis and mooring
deployment/recovery cruises.

**Financial support**

This research has been supported by the National Natural Science Foundation of China (42130407, 42188102).

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
