# Peer review of "Dynamic upper-ocean processes enhance mesopelagic carbon export of zooplankton fecal pellets in the southern South China Sea"

_EGUsphere, 2025_

## Referee Comment (RC2)

**Review Comments**

The study by Wu et al., uses time-series data collected from a sediment trap deployed in the southern South China Sea to provide a year-round assessment of fecal pellet–mediated carbon fluxes and to evaluate how these fluxes respond to upper-ocean dynamics. The findings highlight that, in addition to the seasonal monsoon-driven pattern, other physical processes—such as typhoons, eddies, and episodes of high wind speed—also contribute significantly to the variability of deep fecal pellet carbon fluxes.

Overall, the manuscript is well written and logically organized. The figures are presented effectively, and the results strongly support the main points. I do not have major concerns. My suggestions are mainly minor, relating to figure design, methodological description, and expansion of the discussion to improve clarity and depth. Please see my detailed comments below:

**Specific Comments**

**Line 40**: Please add relevant references after each study approach cited, to demonstrate how these methods have improved our understanding of fecal pellet–mediated carbon fluxes.

**Line 75**: The abbreviation *SCS* has already been defined in a previous section; repetition is unnecessary.

**Lines 165–176**: Cite the corresponding figures or tables when describing results.

**Line 130**: Provide more details (including model data source, configuration, and validation) for the models used to derive the biogeochemical parameters (e.g., Global Ocean Eddy-Resolving Reanalysis, CMEMS Global Ocean Low and Mid Trophic Levels). Many biogeochemical models are available; please justify why these particular models were selected and comment on their performance in the SCS.

**Figure 1**: Consider embedding a small map showing the relative location of the SCS within the broader Pacific Ocean. This would help readers unfamiliar with the region.

**Line 80**: The description that the mixed layer depth (MLD) is "relatively deep" in the SCS may not be appropriate. As shown in your results, there is a significant seasonal cycle, and the maximum MLD remains shallower than 60 m.

**Line 90**: It is difficult to believe that diatoms contribute ~70% of the phytoplankton biomass in the open-ocean upper layer. Do you mean in the sunlit surface water, or are you referring to aggregates/sinking particles in the mesopelagic zone? Please clarify.

**Line 120**: Please add methodological details on how POC flux was quantified from the sediment trap, since the FPC:POC flux ratio is discussed extensively later. In addition, the observation that the FPC:POC ratio remains relatively constant despite increases in total POC flux is intriguing. Does this reflect a true ecological signal— i.e., that different components contributing to POC flux increase proportionally—or could it be an artifact of the methodological approach? Some clarification on this point would strengthen the interpretation.

**Line 125**: The *a* in *Chl a* should be italicized. Please modify throughout the manuscript.

**Line 140**: The *p* in *p-value* should be italicized. Please modify throughout the manuscript.

**Figure 3**: Use different color schemes to denote the two monsoon seasons to improve visualization.

**Figure 5a**: Please clarify which direction the positive values represent.

**Figure 5h**: Report depth-integrated NPP in *mg C m⁻² d⁻¹* to align with the units of POC flux. Also, the time unit is missing.

Precipitation data may be redundant and contribute little to your analysis; consider removing it.

**Figures 6g–6h**: Place surface nitrate on the x-axis, since the aim is to examine how zooplankton respond to upper-ocean dynamics.

**Line 295**: Please clarify how you estimated the ~22-day time lag at your study site.

**Line 335**: Replace *CMD* with the more standard term *deep chlorophyll maximum (DCM)*.

**Line 410**: What is the typical fractional contribution of zooplankton fecal pellets to total POC flux in oligotrophic oceans? Additional discussion and comparison with previous studies would strengthen this section. yibin

---

## Author Comment (AC2)

**Dynamic upper-ocean processes enhance mesopelagic carbon export of zooplankton fecal pellets in the southern South China Sea**

Ruitong Wu1, Zhifei Liu1, Jiaying Li1, Baozhi Lin1, Yulong Zhao1, Junyuan Cao1, and Xiaodong Zhang1 State Key Laboratory of Marine Geology, Tongji University, Shanghai, China

5 Correspondence: Zhifei Liu (lzhifei@tongji.edu.cn)

Abstract. Zooplankton are key contributors to the marine biological pump by converting phytoplankton-derived organic carbon into fast-sinking fecal pellets. Despite the established role of upper ocean dynamics in regulating epipelagic biogeochemistry and plankton communities, their impact on mesopelagic fecal pellet carbon export remains poorly constrained. Here, we present time-series sediment trap mooring observations of fecal pellet fluxes at 500 m from August 2022 to May 2023 in the southern South China Sea. Zooplankton fecal pellet fluxes display distinct seasonal patterns, with average numerical and carbon fluxes of 7.39 × 104 pellets m-2 d-1 and 1.27 mg C m-2 d-1, respectively. Fecal pellets account for 10.0 to 42.6 % (average 21.6 %) of particulate organic carbon export, exceeding most oligotrophic regions. Mesopelagic fecal pellet fluxes are strongly correlated with upper-ocean dynamic processes, including winter mixing, tropical cyclones, and mesoscale eddies. Two tropical cyclones increase regional fecal pellet carbon export by more than 10 % of the annual carbon flux. One spring peak contributes more than 60 % of the total flux, likely driven by the combined effects of winter mixing, cold eddy activity, and spring zooplankton blooms. Our results highlight the critical role of upper-ocean dynamics in fecal pellet carbon export in deep water layers.

**1 Introduction**

The latest Global Carbon Budget2024 unprecedented atmospheric CO2 levels of 422.45 ppma 52 % increase the preindustrial level of 278 ppm, anthropogenic emissions (Friedlingstein et al., 2025). 
[revised manuscript text omitted]

Fecal pellet fluxes at TJ-S exhibit pronounced seasonal variations, with both FPN and FPC showing similar time-series patterns. During August and September, both fluxes were relatively low, reaching minimum values of 937 pellets m-2 d-1 and 0.03 mg C m-2 d-1 mid September, respectively. From October to February, both fluxes steadily increased, peaking at the end of FebruaryFPN flux reached 1.78 × 105 pellets m-2 d-1FPC 2.14 mg C m-2 d-1. peak in late Novemberwhen FPN surg to 8.57 × 104 pellets m-2 d-1, and FPC reach 1.61 mg C m-2 d-1, 4.5 adjacent samples. In early March, fluxes dec, with FPN dropping to 2.98 × 104 pellets m-2 d-1 and FPC decreasing to 0.58 mg C m-2 d-1. However, a sharp increase in both FPN and FPC fluxes in mid March, when FPN surged to 4.61 × 105 pellets m-2 d-1 and FPC 4.62 mg C m-2 d-12 to 3 times greater than the late February andNovember and was 10 to 17 times higher thanadjacent samplesrom April to May, 2.61 × 104 pellets m-2 d-1 and 0.52 mg C m-2 d-1, with a slight increasing trend noted in May.

Fecal pellet contributions revealed distinct contributions to both FPN and FPC flux (Fig. 3c, 3f). The three types of fecal pellets exhibited distinct contributions to both FPN and FPC flux. Ellipsoidal pellets were numerically dominant (3.38 × 104 pellets m-2 d-1, 48 % of total FPN), followed by spherical (2.62 × 104 pellets m-2 d-1, 38%) and cylindrical pellets (1.39 × 104 pellets m-2 d-1, 15 %). During specific seasons, ellipsoidal and spherical pellets together accounted for over, with the first two types collectively representing over 90 % of total FPN. The contribution to carbon flux was size-dependent. Despite numerical

searcity, Cylindrical pellets, despite numerical scarcity, contributed disproportionateld is proportionately to FPC (38 % of total FPC) due to their substantially largester biovolume. Ellipsoidal pellets, though numerically dominant, accounted for 34 % of FPC, while spherical pellets represented only 28 %. The size-driven disparity was particularly pronounced during low-flux periods. For example, in late September, when the total pellet flux reached its minimum, large cylindrical pellets contributed more than 60 % of FPC flux despite their low abundance.

**Table 1.** Geometric parameters and fecal pellet fluxes at TJ-S (500 m) in the southern SCS. Bold values are the average and standard deviations.

| Fecal pellet | Number   | Length                  | Width           | Biovolume                    | FPN flux                                      | FPN            | FPC flux                                | FPC percentage |
|--------------|----------|-------------------------|-----------------|------------------------------|-----------------------------------------------|----------------|-----------------------------------------|----------------|
| type         | measured | (µm)                    | (µm)            | $(\times 10^{6}  \mu m^3)$   | $(\times 10^4 \text{ m}^{-2} \text{ d}^{-1})$ | percentage (%) | (mg C m -2 d -1 ) | (%)            |
| Ellipsoidal  | 1321     | 5 6 –77 6 | 30–422          | 0.03-63.03                   | 0.04-21.08                                    | 37–56          | 0.01-1.53                               | 10–47          |
|              |          | 19 8 ± 118       | 85 + 48  | 1.61 +3.77            | $3.38 \pm 5.70$                               | 48 ±6   | $0.32 \pm \underline{0.43}$             | 34 +7   |
| Cylindrical  | 1167     | 59-3487                 | 20-722          | 0.02-547.92                  | 0.01-9.87                                     | 7–21           | 0.01-2.19                               | 21–59          |
|              |          | 322 + 314        | 84 ± 65  | 6.71 ± 31.01          | $1.39 \pm \underline{2.74}$                   | 15 +4   | $0.38 \pm 0.60$                         | 38 ±10  |
| Spherical    | 1352     |                         | 18–931          | 0.01-422.90                  | 0.04-15.15                                    | 31–46          | 0.01-0.90                               | 19–42          |
|              |          |                         | 104 + 83 | $2.68 \pm \underline{16.08}$ | $2.62 \pm 4.28$                               | 38 +4   | $0.22 \pm 0.26$                         | 28 ± 7  |
| Total        | 3840     | 18-3487                 | 18–931          | 0.01-547.92                  | 0.09-46.10                                    | 100            | 0.03-4.62                               | 100            |
|              |          | 202 + 214        | 92 ± 68  | 3.54 ± 19.81          | 7.39 +12.65                            |                | 0.91 ± 1.27                             |                |

**Fecal pellet flux and constitution (b) FPN flux (x104 pellet m-2 d-1) (a) FPN flux (x104 pellet m-2 d-1) (c) FPN percentage (%) 40 80 30 60 20 10 40 10 20 Aug Sep Oct Nov Dec Jan Feb Mar Apr May Jun Mar Apr Sep Oct Nov Mar (d) FPC flux (mg C m-2 d-1) 2.5 (e) FPC flux (mg C m-2 d-1) (f) FPC percentage (%) 80 60 40 20 Mar Apr Apr May Jun Nov Dec Jan Feb May Mar Apr May Jun Aug Sep Oct Aug Sep Nov Dec Jan Feb ■ Ellipsoidal ■ Cylindrical ■ Spherical → Ellipsoidal → Cylindrical → Spherical ➡ Ellipsoidal ➡ Cylindrical 50 24 (a) FPN (b) FPN (c) FPN (%) 45 60 20 FPN flux (x104 pellet m-2 d-1) 01 g1 05 g2 g2 1 1 1 1 1 1 1 1 1 1 1 1 € 50 percentage ( Nd 20 FPN 10 Aug Sep Oct Nov Dec Jan Feb Mar Apr May Jun Aug Sep Oct Nov Dec Jan Feb Mar Sep Oct Nov Dec Jan Feb Mar Apr May Jun Apr May Jun 2.4 2.2 (e) FPC (d) FPC (f) FPC (%) 4.5 60 2.0 -4.0 (mg C m-2d-1) 9.5 5.5 € 50 percentage ( 3.0 සි 2.5 1.0 Ž 2.0 PC F 0.8 − 1.5 20 1.0 0.4 10 0.5 0.2 Mar Apr May Jun Apr May Jun Oct Nov Dec Jan Feb Aug Sep Oct Nov Dec Jan Feb Mar Aug Sep Oct Nov Dec Jan Feb Mar 2022**

[revised manuscript text omitted]

---

## Author Response (AR1)

**Revision Note on the revised manuscript, dynamic upper-ocean processes enhance mesopelagic carbon export of zooplankton fecal pellets in the southern South China Sea, manuscript no. egusphere-2025-2864, submitted for publication in Biogeosciences.**

We thank all the Editor and two reviewers for their valuable comments and suggestions. We have carefully revised the manuscript in response to all the comments, and we sincerely hope the Editor and referees will be satisfied with our revision. This Revision Note is written based on the annotated (using track changes) version of the manuscript (uploaded in the system). Appended to this letter is our point-to-point response to the comment raised by the reviewers. The notes (in blue) explain how and where each point of comment has been addressed. The line numbers mentioned are new numbers in the annotated version of the manuscript.

**Reviewer#1:**
**'Comment on egusphere-2025-2864', Anonymous Referee #1, 6 Aug 2025**

Fecal pellets act as important particulate organic matter components, and play vital role in marine carbon cycling. This manuscript presented the downward fluxes of zooplankton fecal pellets in the southern South China Sea, based on a long time sediment trap mooring system. The results are very helpful to understand the importance of fecal pellets in organic carbon export and cycling. However, there are some defects through the manuscript. Below are my major advice:

**Reply:** We sincerely appreciate your effort reviewing our manuscript and your positive feedback on our research topic. We have read through all your comments carefully and have made related modifications to improve the quality of our manuscript. We highly appreciate your time and consideration.

**Comment #1:**

The potential impacts of monsoon, typhoon and eddies to fecal pellets export fluxes are discussed. However, through the discussion section, their importance is only qualitatively discussed, and no direct quantitative analysis is done. E.g., in section 4.1, the authors focused on comparing the MLD during monsoon and non-monsoon periods, and then the relationships nitrate and Chl *a* with MLD. I advise the authors quantify the effects of the above events in enhancing the FPN and FPC using statistical analysis/test.

**Reply:** We sincerely thank the reviewer for this valuable and insightful comment. We acknowledge that the initial version of our manuscript primarily provides a qualitative discussion of the effects of the winter monsoon, typhoons and mesoscale eddies on fecal pellet fluxes, without a direct quantitative assessment.

In response, we have now incorporated several additional quantitative approaches in Section 4.4 (Lines 497-519) to better evaluate the influence of these dynamic events on fecal pellet numerical flux (FPN) and fecal pellet carbon flux (FPC).

**1. Correlation among hydrological parameters**

As mentioned in the comment, the relationship between MLD, nitrate, and Chl *a* concentration is intriguing yet not quantified in our manuscript. Thus, we conducted Pearson correlation analyses among key hydrological parameters, including mixed layer depth (MLD), sea level anomaly (SLA), sea surface temperature (SST), zooplankton biomass (ZMC), and calculated all correlation matrix. The results are presented in Figure 9 in Line 513.

**2. Relationships between fecal pellet fluxes and hydrological parameters**

In addition, we applied Mantel tests to further access the potential relationships between fecal pellet fluxes (FPN, FPC) and hydrological parameters. We found significant correlations between fecal pellets and SLA, ZMC, and other components of the BCP, such as opal, carbonate, and terrigenous fluxes. These results reinforce our previous hypothesis that upper-ocean dynamics strongly modulate particle export, and that various BCP components tend to increase simultaneously under such influences. The results, also shown in Figure 9, are discussed in Lines 498-504.

**3. General linear model (GLM):**

To quantify the relative contributions of upper-ocean dynamic events, we employed a general linear model (GLM). Our results indicate that The East Asian monsoon (EAM) exerts a demonstrated positive influence, explaining approximately 42 % of FPN and over 32 % of FPC variability. The model outputs are summarized in Table 2 (Line 516), with interpretations provided in Lines 505-512. However, we note that the limited sample size (n = 13) may constrain model robustness and introduce potential overfitting, which we discuss in Lines 510-512, emphasizing the need for future studies with larger datasets.

**Comment #2:**

The language and logic of the manuscript must be improved before submitting and published to such high quality journal.

**Reply:** We sincerely thank the reviewer for this valuable suggestion. To meet the standards of such a high-quality journal, we have made comprehensive improvements to both the **language** and **logic structure** of the manuscript.

**1. Language:**

We carefully revised the entire manuscript to enhance readability and academic tone. The language was refined to be more professional and concise, with redundant expressions removed. We also adjusted sentence structures to improve clarity and coherence. Since revisions were made throughout the text, we provide one sample for illustration:

Previous version (Section 2.1):

'*The South China Sea (SCS), the largest semi-enclosed marginal sea in the western Pacific, spans $3.5 \times 10^6$ km² with an average depth of 1140 m (Wang and Li, 2009).*

*Our study focuses on the southern SCS near the Sunda Shelf (Fig. 1a), where the East Asian monsoon system (EAM) dominates surface wind fields and ocean circulation (Shaw and Chao, 1994).'*

Revised version (Lines 81-85):
*'Mooring TJ-S (6.72ºN, 110.76ºE, 1630 m water depth) is located in the southern SCS near the Sunda Shelf (Fig. 1a), where the East Asian monsoon system (EAM) exerts dominant control on local wind fields and ocean circulation (Shaw and Chao, 1994).'*

This revision removed redundant background information and strengthens focus on the study site.

**2. Logic:**
We also reorganized sections of the manuscript to improve logical flow and strengthen discussions. We now improved the logic of BCP introduction (Lines 19-29), refined the description of methodological approaches (Lines 33-41), and enhanced the logical flow in describing hydrological data sources (Lines 144-175). We have now strengthened the logic in the introductory part of the discussion section (Lines 296-323), and we also reorganized paragraphs and removed redundant information in the further discussion to maintain a clear focus on major arguments (Lines 325-338, 396-402).

**Comment #3:**
The manuscript only simply presented the data sources, such as wind field, primary production…. But, how the trap system pretreated before using? In other words, how the bottles cleaned and did any chemicals (e.g. $HgCl_2$ or NaCl solution) added to each bottle? Since the storing durations of the particles in the bottles are different and the degradation percent in each bottle should be considered. In addition, POC is used in Figure 4, but how was it analyzed is not mentioned in the method section.

**Reply:** We thank the reviewer for this valuable comment. In response, we have expanded Section 2.2 (Lines 111-142) to include detailed descriptions of the pretreatment and handling of the sediment trap system as well as procedures for POC measurement. Specifically, we have added information on the use of NaCl-buffered $HgCl_2$ solution prior to deployment as a preservative to minimize biological degradation during sample storage (Lines 114-117). We also added detailed steps involved in sample processing after retrieval, including sample storage conditions, rinsing and splitting of subsamples, and drying procedures (Lines 122-127). The analytical method for total mass flux (TMF) and POC determination are also described in Lines 128-130.

**Comment #4:**
There are some mistakes/typos in certain figures: There is no panel d in Figure 4. The y-axis names are repeated in Figure S7, the right one should be wrong. Base on the contents, Figure 3a and Figure S7d are repeated. Figure 3a and 3b, 3d and 3e are also repeated. Figure 5 and Figure 7 are partially repeated and the logic/order in Figure 5 are bad (they should be presented

in the order of physical parameters, then biochemical parmeters, and raw data and then reanalized data). Additionally, the panels should be cited for the first time in the main text sequentially.

**Reply:** We thank the reviewer for this careful examination and for pointing out the errors in our figures. In response, we have thoroughly checked and revised all the figures in both the main text and the supplementary materials: Figure 3 has been reorganized to avoid data repetition: Fig. 3a now presents the total FPN flux, while Fig. 3b now shows the individual contributions of different fecal pellet types (Line 237). All panels in Figure 4 have been carefully reviewed and corrected (Line 256). The orders of hydrological parameters in Figure 5 have been completely reorganized to improve logic flow. As suggested, physical parameters and raw data are now presented first, followed by reanalyzed data and biogeochemical parameters (Line 289). In Figure S7, panel d has been removed to eliminate redundancy (Line 110, supplementary materials). The right y-axis is now correctly labeled as representing the mean biovolume of the single pellet (the line plot), while the left y-axis represents the biovolume distribution (bar plot). Additionally, we have ensured that all panels are now cited sequentially in the main text according to their first appearance.

**Comment #5:**
Please add note for the bold numbers in Table 1. I think they the average values. Please also present the standard deviations.

**Reply:** We thank the reviewer for this helpful suggestion. We have now clarified in the caption of Table 1 (Lines 234-235) that the bold numbers represent average values. In addition, we have now included the corresponding standard deviations to provide a more comprehensive statistical description.

**Comment #6:**
Lines 25-26: "sequestering over 1300 Pg C" Is this the integrating value? If yes, what's the time scale?

**Reply:** We thank the reviewer for pointing out this issue. The 1300 Pg C value is assimilated by an ensemble numerical model (Nowicki et al., 2022), and it is an **integrating value** with an average sequestration time of $127^{133}_{122}$ years. Detailed calculation methodology is described in section 2.6.2 of their research by using the equation below:

$$\frac{dc}{dt} = \mathbf{A}c + \mathbf{J}_{DIC} - \mathbf{Q}_{surf}c$$

where $\mathbf{A}$ is the transport matrix and c is DIC from respired organic matter. $\mathbf{J}_{DIC}$ is the source of DIC from organic matter respiration, and $\mathbf{Q}_{surf}$ is a matrix operator that removes carbon from the system once it reaches the surface layer. $\mathbf{J}_{DIC}$ is obtained by solving the model equations for each carbon sequestration pathway in the same way as the export pathways are partitioned, using a diagnostic linear model to avoid issues associated with nonlinearities in the original model equations (Nowicki et al., 2022).

We have realized that using this number may be not proper to merely demonstrate the importance of BCP in this paragraph. We have now reorganized the whole paragraph into: "*The ocean plays a pivotal role in regulating global carbon sink, absorbing 2.9 ± 0.4 Gt C annually through coupled physical and biological mechanisms, mitigating increasing anthropogenic carbon dioxide ($CO_2$) emissions (Friedlingstein et al., 2025). Central to this uptake lies the biological carbon pump (BCP), which converts massive dissolved $CO_2$ in the surface ocean into particulate organic carbon (POC) via phytoplankton photosynthesis (Falkowski, 2012; Boyd and Trull, 2007; Nowicki et al., 2022).*" (Line 20-24)

**Comment #7:**
Lines 37-38: The authors mentioned "These studies", but did not cite any references, please add the relevant references. In addition, why the contribution of fecal pellets to POC can > 100%?

**Reply:** We thank the reviewer for the careful examination and for pointing out the missing citations. Accordingly, we have now added relevant references after this sentence (Lines 33-41), including *Shatova et al., 2012; Siegel et al., 2014; Turner et al., 2015; Stamieszkin et al., 2015; Estapa et al., 2017; Li et al., 2022; Countryman et al., 2022; Terrats et al., 2023; Darnis et al., 2024.*

Regarding the FPC/POC ratio occasionally exceeding 100 %, this phenomenon arises primarily from methodological rather than ecological causes. Theoretically, FPC is a component of total POC, thus the actual FPC/POC ratio should not surpass 100 %. However, FPC and POC are quantified using different approaches. POC is directly measured from bulk samples, while FPC flux is estimated from pellet volumes converted to carbon contents using carbon-volume conversion factors. Uncertainties in pellet size measurement, image recognition, and the choice of conversion factors can lead to over estimations of FPC. In some cases, especially when fecal pellets dominate the local BCP process, these uncertainties may produce calculated FPC/POC ratios close to or slightly above 100 %, though the true ratio cannot exceed this limit.

**Comment #8:**
Line 110: Based on the location of the mooring station and depth, terrigenous particles may be not the only component of the fine particles. I mean biological originated organic/inorganic particles, including detritus of phytoplankton, partially degraded organic particles, etc, may contribute more to this fraction.

**Reply:** We thank the reviewer for raising this insightful comment. We fully agree that, given the location and depth of the TJ-S mooring station, the fine-particle fraction in our sediment trap samples likely includes not only terrigenous materials, but also biologically derived components such as phytoplankton detritus and partially-degraded organic and inorganic materials. We apologize for not clearly describing all the relevant pretreatment procedures in the original version.

In the revised manuscript, we have expanded Section 2.2 (Lines 122-130) to include more

details on sample pretreatment, processing, and the procedures used to separate FPs from other components, thereby clarifying the composition and handling of the fine-particle fraction.

At station TJ-S, our retrieved trap samples were divided into terrigenous particles, opal, carbonate, and organic carbon. The total mass flux (TMF) was calculated from sample dry weight (mg) normalized to the trap collection area and sampling duration, while POC and component-specific mass fluxes (including opal flux, carbonate flux, and terrigenous flux) were derived by the measured TMF percentage (%). Before zooplankton numeration, these fluxes were removed to ensure accurate quantification of fecal pellet contributions.

**Comment #9:**
Lines 155-156: Why the authors say spherical pellets are "slightly smaller"? What's the criteria? Based on the average biovolume, it's much smaller than cylindrical ones, but larger than the ellipsoidal ones (not similar). Please do statistical significance and present p values.

**Reply:** We thank the reviewer for the careful examination and for pointing out this problem. We agree that, based on the average biovolume data, our previous statement was not precise.

In response, we have now conducted ANOVA tests to compare the biovolumes of fecal pellets with different shapes. The results indicate that cylindrical pellets were significantly larger than both elliptical ($p < 0.001$) and spherical ($p < 0.001$) pellets, while spherical pellets were also larger than elliptical pellets ($p < 0.05$). We have now corrected the expressions and added these results in our revised manuscript (Lines 192-194).

**Comment #10:**
Lines 176-178: I am confused here. The contributions of ellipsoidal, spherical and cylindrical to FPN are 48%, 38% and 15%, respectively, so how can we got > 90% for ellipsoidal + spherical?

**Reply:** We thank the reviewer for the careful examination and for pointing out the inconsistency. We apologize for the inaccurate expression in the original text. The values of 48 %, 38 % and 15 % represent annual averages, whereas during specific periods, the combined contribution of ellipsoidal and spherical pellets indeed exceeded 90 % (e.g., 93 % in UP01 during September, and 92 % in UP08 during January). To clarify this point and avoid confusion, we have revised this sentence into: '*During specific seasons, ellipsoidal and spherical pellets together accounted for over 90 % of total FPN.*' (Lines 227-228)

**Comment #11:**
Lines 244-249: What's the relationship of these sentences and pellets export? I do not think they are necessary in discuss how phytoplankton blooms. I also suggest to re-organize this section to make it simple and clear.

**Reply:** We thank the reviewer for this constructive suggestion. We agree that, the discussions of the plankton bloom hypothesis was not directly relevant to fecal pellet export, and the logic

of this section required improvement. In response, we have now removed redundant information unrelated to fecal pellet export (e.g., general theories of phytoplankton blooms), and reorganized this section to improve clarity, coherence and focus on the topic. (Lines 296-323)

**Comment #12:**
Line 252: "microbial remineralization and degradation", please delete "and degradation", because remineralization is generally equal to degradation.

**Reply:** We thank the reviewer for the careful examination. We agree with this suggestion, and we have now uniformly use the term 'remineralization' instead of 'degradation' in our revised manuscript. The revised sentence reads: '*Processes during sedimentation, such as microbial remineralization, can strongly regulate the transform efficiency but remain difficult to quantify due to the lack of downward sediment trap samples.*' (Lines 311-312)

**Reviewer#2:**
**'Comment on egusphere-2025-2864', Anonymous Referee #2, 14 Sep 2025**

The study by Wu et al., uses time-series data collected from a sediment trap deployed in the southern South China Sea to provide a year-round assessment of fecal pellet–mediated carbon fluxes and to evaluate how these fluxes respond to upper-ocean dynamics. The findings highlight that, in addition to the seasonal monsoon-driven pattern, other physical processes— such as typhoons, eddies, and episodes of high wind speed—also contribute significantly to the variability of deep fecal pellet carbon fluxes.

Overall, the manuscript is well written and logically organized. The figures are presented effectively, and the results strongly support the main points. I do not have major concerns. My suggestions are mainly minor, relating to figure design, methodological description, and expansion of the discussion to improve clarity and depth. Please see my detailed comments below:

**Reply:** We sincerely appreciate your effort reviewing our manuscript and your positive feedback on the manuscript. We have read through all your comments carefully and have made related modifications to improve the quality of our manuscript. We highly appreciate your time and consideration.

**Specific Comments:**
**Comment #1:**
Line 40: Please add relevant references after each study approach cited, to demonstrate how these methods have improved our understanding of fecal pellet–mediated carbon fluxes.

**Reply:** We thank the reviewer for this helpful suggestion. Accordingly, we have added the relevant references to support each study approach mentioned to strengthen this statement

(Lines 33-41). We now have included references for study approaches including in situ observations (*Shatova et al., 2012; Turner et al., 2015; Li et al., 2022; Wang et al., 2023; Cao et al., 2024; Darnis et al., 2024*), satellite observations (*Siegel et al., 2014*), Bio-Argo float profiling (*Estapa et al., 2017; Terrats et al., 2023*), and numerical modeling (*Stamieszkin et al., 2015; Countryman et al., 2022*).

**Comment #2:**
Line 75: The abbreviation SCS has already been defined in a previous section; repetition is unnecessary.

**Reply:** We thank the reviewer for this helpful comment. We agree that this abbreviation is already discussed and is not necessarily presented. To avoid unnecessary repetition, we have now revised Section 2.1 by removing the redundant abbreviation of SCS and replacing it with a clearer description of the mooring station TJ-S (Line 81-85).

The revised text now reads: *'Mooring TJ-S (6.72ºN, 110.76ºE, 1630 m water depth) is located in the southern SCS near the Sunda Shelf (Fig. 1a), where the East Asian monsoon system (EAM) exerts dominant control on local wind fields and ocean circulation (Shaw and Chao, 1994).'*

**Comment #3:**
Lines 165–176: Cite the corresponding figures or tables when describing results.

**Reply:** We thank the reviewer for this helpful suggestion. We apologize for not citing corresponding figures or tables in our discussion sections. In response, we have revised the Results sections and added citations to corresponding figures and tables to improve clarity (Line 188-209).

**Comment #4:**
Line 130: Provide more details (including model data source, configuration, and validation) for the models used to derive the biogeochemical parameters (e.g., Global Ocean Eddy-Resolving Reanalysis, CMEMS Global Ocean Low and Mid Trophic Levels). Many biogeochemical models are available; please justify why these particular models were selected and comment on their performance in the SCS.

**Reply:** We thank the reviewer for this valuable suggestion. In response, we have expanded Section 2.3 to include additional details on the model and the data sources, including their configuration, data assimilation schemes, and their reported performance in the South China Sea (Line 144-175). In addition, we have added a brief justification for selecting these particular products and cited recent studies that validate or employ them in this region (Line 146-148).

**Comment #5:**
Figure 1: Consider embedding a small map showing the relative location of the SCS within the broader Pacific Ocean. This would help readers unfamiliar with the region.

**Reply:** Thank you for this helpful advice. We have now added a small global map showing the relative location of SCS in Figure 1 (Fig. 1b, Line 104).

**Comment #6:**
Line 80: The description that the mixed layer depth (MLD) is "relatively deep" in the SCS may not be appropriate. As shown in your results, there is a significant seasonal cycle, and the maximum MLD remains shallower than 60 m.

**Reply:** We thank the reviewer for this valuable advice. We agree with your concern that our previous description is not appropriate. To provide a more accurate summary of MLD variability in the southern SCS, we added relevant references (*Qu et al., 2007; Thompson and Tkalich, 2014; Liang et al., 2019*) and have rewrote this sentence accordingly. The new version reads:
'*In this region, the mixed layer depth (MLD) is primarily controlled by air-sea heat fluxes and exhibits remarkable seasonal signals under monsoon forcing, deepening during June-August and December-February but remaining shallower than 60 m annually.*' (Line 90-93)

**Comment #7:**
Line 90: It is difficult to believe that diatoms contribute ~70% of the phytoplankton biomass in the open-ocean upper layer. Do you mean in the sunlit surface water, or are you referring to aggregates/sinking particles in the mesopelagic zone? Please clarify.

**Reply:** We thank the reviewer for pointing out this issue. We agree with your concern. The previously cited values refer to **sunlit surface water** contributions, and we apologize for the inaccurate expression in the original version. In response, we have added clarifications and relevant references (Lines 96-99) and revised the sentence as follows: '*Diatoms and dinoflagellates dominate the surface phytoplankton community in the region, with contributions from picophytoplankton and cyanobacteria (Zhu et al., 2003; Ke et al., 2012, 2016; Wang et al., 2022).*'

**Comment #8:**
Line 120: Please add methodological details on how POC flux was quantified from the sediment trap, since the FPC:POC flux ratio is discussed extensively later. In addition, the observation that the FPC:POC ratio remains relatively constant despite increases in total POC flux is intriguing. Does this reflect a true ecological signal—i.e., that different components contributing to POC flux increase proportionally—or could it be an artifact of the methodological approach? Some clarification on this point would strengthen the interpretation.

**Reply:** We thank the reviewer for this valuable comment. In response, we have expanded Section 2.2 to include additional methodological details on how POC flux was quantified from the sediment trap (Line 126-130). Since POC flux and FPC flux are calculated independently, we infer that the relative constant FPC/POC ratio reflects a true ecological signal rather than methodological artifact, suggesting that different components contributing to POC flux increased proportionally during periods of increased pellet export.

**Comment #9:**
Line 125: The a in Chl a should be italicized. Please modify throughout the manuscript.
**Comment #10:**
Line 140: The p in p-value should be italicized. Please modify throughout the manuscript.

**Reply:** Thank you for this advice. We have corrected the term Chl *a* and *p*-value in the manuscript as suggested, and have ensured its consistency throughout the text. Specific corrections are listed as follows:

Chl *a*: Line 159, Line 345, Line 353, Line 445, Line 471, Line 488, Line 494.
*p* values: Line 180, Line 194, Line 195, Line 342, Line 343, Line 347, Line 348, Line 500, Line 501, Line 507.

**Comment #11:**
Figure 3: Use different color schemes to denote the two monsoon seasons to improve visualization.

**Reply:** Thank you for this advice. We have changed the color scheme of summer and winter monsoon in Figure 3 (Line 237), Figure 4a, 4b (Line 256), Figure 5 (Line 288), Figure 7k (Line 443), and Figure 8j (Line 492) to improve visualization.

**Comment #12:**
Figure 5a: Please clarify which direction the positive values represent.
**Reply:** We thank the reviewer for this helpful comment. We apologize for not adding clarifications of wind directions in the caption of Figure 5. Fig. 5a is the eastward component of 10 m wind, where positive values represent eastern winds and negative values for western winds. Similarly, Fig. 5b is the northward component, where positive values represent northern winds, and negative values for southern winds. To improve clarification, we have now revised the description of Figure 5 as follows: '*(a) eastward component for 10 m wind (u), where positive values represent eastern winds, and negative values for western wind; (b) northward component for 10 m wind (v), where positive values represent northern winds, and negative values for southern winds.*' (Lines 289-291)

**Comment #13:**
Figure 5h: Report depth-integrated NPP in mg C m$^{-2}$ d$^{-1}$ to align with the units of POC flux. Also, the time unit is missing.
Precipitation data may be redundant and contribute little to your analysis; consider removing it.

**Reply:** Thank you for this helpful advice. We have changed the units of NPP in Figure 5j to align with POC flux, and added corresponding time units. We agree that the precipitation data is not used in our further discussion and analysis, and this data has been removed (Figure 5, Line 288).

**Comment #14:**

Figures 6g–6h: Place surface nitrate on the x-axis, since the aim is to examine how zooplankton respond to upper-ocean dynamics.

**Reply:** Thank you for this helpful advice. We agree that in our discussion, zooplankton biomass should be the dependent variable (y axis), since we aim to examine how they correspond to upper-ocean dynamics. Thus, the nutrients should be in x axis, as they represent independent variables. In response, we have revised Figure 6 and changed the x-axis of Figures 6a-6d to examine the respond of biochemical parameters to upper ocean dynamics. (Line 350)

**Comment #15:**

Line 295: Please clarify how you estimated the ~22-day time lag at your study site.

**Reply:** We thank the reviewer for this valuable comment. The estimated time lag between primary production and fecal pellet export at 500 m was inferred mainly from previous studies in the SCS (e.g., Bao et al., 2023), where biological responses were reported to occur within several days to weeks. In our study, each sediment-trap sample integrated over a 22-day period; therefore, we assumed that events occurring during one sampling interval (e.g., UP06) could influence both that sample and the subsequent one (e.g., UP07). This assumption guided our interpretation that WSP 2 and WSP 3 may correspond to PK 2 and PK 3. However, due to the lack of concurrent surface observations (e.g., phytoplankton or zooplankton analyses), the exact time lag at our station cannot be determined quantitatively.

**Comment #16:**

Line 335: Replace CMD with the more standard term deep chlorophyll maximum (DCM).

**Reply:** Thank you for this helpful advice. We have corrected the term deep chlorophyll maximum (DCM) in the manuscript as suggested, and have ensured its consistency throughout the text and figures.
Specific corrections are listed as follows:
DCM: Line 174, Line 293, Line 359, Line 362, Line 434, Line 446, Line 494, Line 525.

**Comment #17:**

Line 410: What is the typical fractional contribution of zooplankton fecal pellets to total POC flux in oligotrophic oceans? Additional discussion and comparison with previous studies would strengthen this section.

**Reply:** Thank you for this valuable comment. We agree with that including comparative discussion will strengthen our interpretation. Accordingly, we have reorganized this sentence as follows:
*'In the SCS, zooplankton fecal pellets make the most contribution to POC export in the southern region, with FPC/POC ratio ranging from 10.0 % to 42.6 %, reaching an average of 21.6 %. This value is larger than most oligotrophic regions including the central north Pacific (Wilson et al., 2008), the Sargasso Sea (Shatova et al., 2012), and the Mediterranean (Carroll et al.,*

[revised manuscript text omitted]

---

## Referee Report (RR1)

The authors made great efforts in improving the manuscript. Now the revised manuscript is good in science, logic and other details. It can be accepted after some minor revision. The detailed comments are below:

1. Line 21: Please change "which converts massive, dissolved CO2.." to "which convert massive CO2 …"

2. Line 163: Figure 3 should be Fig. 3

3. Please add units for the mean values in lines 168-169.

4. I still think Figures 3a and b, d and e are overlapped, and Figures 3b and e were not cited in the text. I suggest present the total FPN and three kinds of FPN in one panel, FPC in the same manner.

5. The legends are not clear for Figure 4b. There are four lines in this panel, but the authors did not explain the meaning of each line. In addition, Figure 4c was not cited in the text. In line 205, the authors said FPC/POC displayed inverse seasonal variation to POC fluxes based on Figure 4b, but according to Figure 4c, positive correlations between POC flux to FPN and FPC. To some extent, it seems contradictory.

6. In figure 6, the points are shown in different colors, please using color bar or other means to explain them. Also the dashed lines are in red and blue, which one denotes EAM and non-EAM?

---

## Author Response (AR2)

**Revision Note on the revised manuscript, dynamic upper-ocean processes enhance mesopelagic carbon export of zooplankton fecal pellets in the southern South China Sea, manuscript no. egusphere-2025-2864, submitted for publication in Biogeosciences.**

We thank all the reviewers for their valuable comments and suggestions. We have carefully revised the manuscript in response to all the comments, and we sincerely hope the editor and referees will be satisfied with our revision. This Revision Note is written based on the annotated (using track changes) version of the manuscript (uploaded in the system). Appended to this letter is our point-to-point response to the comment raised by the reviewers. The notes (in blue) explain how and where each point of comment has been addressed. The line numbers mentioned are new numbers in the annotated version of the manuscript.

**Reviewer#1: 'Comment on egusphere-2025-2864', Anonymous Referee #1, 24 Nov 2025**
The authors made great efforts in improving the manuscript. Now the revised manuscript is good in science, logic and other details. It can be accepted after some minor revisions. The detailed comments are below:

**Reply:** We sincerely appreciate the reviewer's time and effort in evaluating our revised manuscript. We are grateful for your positive comments regarding the scientific quality, logical structure and improvements. We have carefully addressed all minor comments and have incorporated corresponding changes throughout the manuscript. Thank you again for your constructive feedback and valuable guidance which have largely strengthened the quality of our work.

**Comment #1:**
Line 21: Please change "which converts massive, dissolved CO2.." to "which convert massive CO2 …"

**Reply:** Thank you for this helpful comment. We agree that our previous expression "dissolved CO2" was not accurate and could lead to misunderstandings of the BCP carbon sources, as surface ocean inorganic carbon pool includes multiple species, not only dissolved $CO_2$.

Accordingly, we have now corrected this sentence as follows: *"Central to this uptake lies the biological carbon pump (BCP), which converts massive $CO_2$ in the surface ocean into particulate organic carbon (POC) via phytoplankton photosynthesis (Falkowski, 2012; Boyd and Trull, 2007; Nowicki et al., 2022)."* (Lines 21-23)

**Comment #2:**
Line 163: Figure 3 should be Fig. 3

**Reply:** Thank you for this helpful comment. We have now examined all figure citations in our manuscript to ensure consistency, and have corrected this sentence as follows: *"Geometric and flux characteristics are summarized in Table 1 and Fig. 3, with detailed data in Table S1."* (Lines 167-168)

**Comment #3:**
Please add units for the mean values in lines 168-169.

**Reply:** Thank you for this helpful comment. We have added units for mean values and checked the unit consistency throughout the manuscript. The revised sentence now reads: *"FPN ranged from $9.4 \times 10^2$ to $4.61 \times 10^5$ pellets $m^{-2}$ $d^{-1}$ (mean: $7.39 \times 10^4$ pellets $m^{-2}$ $d^{-1}$, Fig. 3a), while FPC spanned from 0.03 mg C $m^{-2}$ $d^{-1}$ to 4.62 mg C $m^{-2}$ $d^{-1}$ (mean: 0.91 mg C $m^{-2}$ $d^{-1}$, Fig. 3d), both exhibiting pronounced seasonal variations."* (Lines 173-174)

**Comment #4:**
I still think Figures 3a and b, d and e are overlapped, and Figures 3b and e were not cited in the text. I suggest present the total FPN and three kinds of FPN in one panel, FPC in the same manner.

**Reply:** Thank you for this helpful comment. Indeed, the datasets of Figures 3a-b, d-e were identical and were often discussed together in the text, which further resulted in data redundancy of different subplots and missing citations of figures in the manuscript.

We have carefully evaluated your suggestion of combining total and component pellet fluxes into a single panel. Across most plotting methods, we think stacked bar charts remain the most effective way of visualizing our data, as they can display both total fluxes and contributions of individual FP types, while preserving the seasonal patterns (as in the original Fig. 3a and d).

In response, we have now removed Fig. 3b and 3e to avoid data overlapping and have reorganized the sequence of subplots according to their reference in the text. The revised Figure 3 now consists of 4 subplots, including (a) FPN flux (stacked bar chart showing total FPN and fluxes of three FP types), (b) FPN percentage (%); (c) FPC flux (stacked bar chart showing total FPC and fluxes of three FP types); and (d) FPC percentage (%). The captions and references of the figure have been updated accordingly. (Lines 200-203).

**Comment #5:**
The legends are not clear for Figure 4b. There are four lines in this panel, but the authors did not explain the meaning of each line. In addition, Figure 4c was not cited in the text. In line 205, the authors said FPC/POC displayed inverse seasonal variation to POC fluxes based on Figure 4b, but according to Figure 4c, positive correlations between POC flux to FPN and FPC. To some extent, it seems contradictory.

**Reply:** Thank you for this helpful comment. We sincerely apologize for the unclear legends in Figure 4b and the confusion caused by the incomplete citation in Figure 4c. In response, we have now redrawn and reorganized Figure 4 with clearer legends and descriptions (Lines 218-220). Specifically, Figure 4c now illustrates the correlations among FPN, FPC and POC flux, while a new panel (Fig. 4d) has been added to show the relationship between the FPC/POC ratio and POC flux.

As shown in Figure 4c, FPN, FPC, and POC fluxes are all positively correlated with each other, with all linear relationships being statistically significant ($p < 0.01$). However, though FPC and POC flux are positively correlated, their ratio does not scale with POC flux and can display opposite seasonal patterns (Fig. 4d). For example, between August and November 2022, POC fluxes remained extremely low, whereas the FPC/POC ratio reached its annual maximum (Fig. 4b).

**Comment #6:**
In figure 6, the points are shown in different colors, please using color bar or other means to explain them. Also the dashed lines are in red and blue, which one denotes EAM and non-EAM?

**Reply:** Thank you for this helpful advice. We sincerely apologize for the unclear visualization in the previous version of Figure 6. The blue-green-yellow color range previously used in the scatter plot did not correspond to any specific meaning and may have caused confusion. In response, we have now redrawn and reorganized Figure 6 for clearer interpretation. In the revised plot, the two groups of data are distinctly colored, with EAM in blue and non-EAM in red, and the linear regression lines are shown in the same corresponding colors. (Line 273)

**Reviewer#2: 'Comment on egusphere-2025-2864', Anonymous Referee #2, 24 Nov 2025**

**Comment #1:**
Line 35: The studies by Estapa et al. (2017) and Terrats et al. (2023) provide estimates of carbon fluxes associated with both large and small particles. However, we do not think that the "large-particle flux" in these studies can be directly interpreted as fecal pellet–driven carbon fluxes.

**Reply:** Thank you for this helpful comment. We agree that the particle fluxes reported in Estapa et al. (2017) and Terrats et al. (2023) cannot be directly interpreted as fecal pellet carbon fluxes, and therefore these studies are not suitable to support the sentence. In the previous manuscript, they were cited as evidence for the use of Argo in fecal pellet research. However, Argo and Bio-Argo estimates also include both small and large particles, and the contribution of fecal pellets cannot be separated.

In response, we have now removed the two references and revised the sentence as follows: *"In-situ observations from sediment traps and large filtering systems provide high-resolution time-series flux records (Shatova et al., 2012; Turner et al., 2015; Li et al., 2022; Wang et al., 2023; Cao et al., 2024; Darnis et al., 2024), while complementary approaches including satellite observations (Siegel et al., 2014) and numerical modeling (Stamieszkin et al., 2015; Countryman et al., 2022) largely expand the scope of investigation across broader spatial and temporal scales."* (Lines 33-38)

**Comment #2:**
Line 325: The abbreviation "EAM" has been repeatedly defined throughout the manuscript. It should be introduced only once when it first appears.

**Reply:** Thank you for this helpful comment. We have now examined all terms, definitions, and abbreviations throughout the manuscript, including EAM, BCP, POC, SCS, WSPs, TCs, and CEs, and ensured that each term is introduced only once at its first occurrence. The specific first appearances are as follows: EAM (Line 62), BCP (Line 21), POC (Line 22), SCS (Line 61), WSPs (Line 227), TCs (Line 228), CEs (Line 329). Thank you again for your careful examination.

**Comment #3:**
Table 2: It is somewhat unclear how the generalized linear model (GLM) was used to quantify the relative contribution of physical events to carbon export. For each event type, was it treated as a single independent factor or as a composite factor including multiple environmental variables? Please provide more methodological details to clarify this.

**Reply:** Thank you for this helpful comment. In our GLM model, each physical event (typhoon, monsoon, eddy) was treated as an independent binary factor, coded as 1 when the event occurred and 0 otherwise. Since the aim of this analysis was to evaluate the relative contribution of each event to fecal pellet export, we did not combine multiple environmental variables into composite factors, as doing so could increase the model uncertainty and the risk of overfitting, giving the limited number of observations we have (n = 13).

In response, we have expanded Section 2.4 to include more description of the GLM model. The revised text reads as follows:
*"To evaluate the relative contribution of different events to fecal pellet export, we applied a generalized linear model (GLM) with a Gamma distribution and a log link function:*

$$FPN \sim Monsoon + Typhoon + Eddy$$

*Each physical event was treated as an independent binary factor indicating whether the event occurred during the sampling period."* (Lines 157-162)

**Comment #4:**

Line 530: The typical range of the FPC/POC ratio in oligotrophic oceans should be specified, along with the corresponding references.

**Reply:** Thank you for this helpful comment. We agree that specifying the range of FPC/POC ratio in oligotrophic oceans is essential and can greatly strengthen our discussion. According to recent studies and earlier reviews, FPC/POC ratio ranges from 0.3 % to 35 % in oligotrophic mesopelagic (200–1000 m) regions. Reported values and related references include: the southern SCS (1.3–30.0 % at 500 m, mean 9.6 %; Li et al., 2022); the northern SCS (0.3–15.7 % at 500 m, mean 3.4 %; Wang et al., 2023); the western SCS (0.7–28.2 % at 500 m, mean 4.4 %; Cao et al., 2024); Shikoku, Japan (0.4–1.7 % at 500 m; Ayukai and Hattori, 1992); the central North Pacific (14–35 % at 500 m; Wilson et al., 2008); the northwestern Mediterranean (3–35 % at 500 m; Carroll et al., 1998), and the Sargasso Sea (0.4–10.0 % at 500 m; Shatova et al., 2012).

In response, we have now revised the sentences as follows:
*"In the SCS, zooplankton fecal pellets make the most contribution to POC export in the southern region, with FPC/POC ratio ranging from 10.0 % to 42.6 %, reaching an average of 21.6 %. This range is relatively higher than the typical values reported for oligotrophic mesopelagic regions (0.3–35 %; reviewed in Turner et al., 2015; Li et al., 2022), including Station ALOHA in the central North Pacific (14–35 %; Wilson et al., 2008), the northwestern Mediterranean (3–35 %; Carroll et al., 1998), and the Sargasso Sea (0.4–10.0 % at 500 m; Shatova et al., 2012), highlighting the critical role of zooplankton fecal pellets in shaping the unique carbon export process in the southern SCS."* (Lines 401-407)

**References cited in this Reply:**

Ayukai, T., & Hattori, H.: Production and downward flux of zooplankton fecal pellets in the anticyclonic gyre off Shikoku, Japan, Oceanolog. Acta, 15(2), 163-172, 1992.

Bao, M., Xiao, W., and Huang, B.: Progress on the time lag between marine primary production and export production, J. Xiamen Univ. (Nat. Sci.), 62, 314–324, https://doi.org/10.6043/j.issn.0438-0479.202204044, 2023.

Carroll, M. L., Miquel, J. C., and Fowler, S. W.: Seasonal patterns and depth-specific trends of zooplankton fecal pellet fluxes in the Northwestern Mediterranean Sea, Deep Sea Res. Part I, 45, 1303–1318, https://doi.org/10.1016/s0967-0637(98)00013-2, 1998.

Cao, J., Liu, Z., Lin, B., Zhao, Y., Li, J., Wang, H., Zhang, X., Zhang, J., and Song, H.: Temporal and vertical variations in carbon flux and export of zooplankton fecal pellets in the western South China Sea, Deep Sea Res. Part I, 207, 104283, https://doi.org/10.1016/j.dsr.2024.104283, 2024.

Countryman, C. E., Steinberg, D. K., and Burd, A. B.: Modelling the effects of copepod diel vertical migration and community structure on ocean carbon flux using an agent-based model, Ecol. Modell., 470, 110003, https://doi.org/10.1016/j.ecolmodel.2022.110003, 2022.

Darnis, G., Geoffroy, M., Daase, M., Lalande, C., Søreide, J.E., Leu, E., Renaud, P.E. and Berge, J.: Zooplankton fecal pellet flux drives the biological carbon pump during the

winter–spring transition in a high-Arctic system, Limnol. Oceanogr., 69, 1481–1493, https://doi.org/10.1002/lno.12588, 2024.

Estapa, M., Durkin, C., Buesseler, K., Johnson, R., and Feen, M.: Carbon flux from bio-optical profiling floats: calibrating transmissometers for use as optical sediment traps, Deep Sea Res. Part I., 120, 100–111, https://doi.org/10.1016/j.dsr.2016.12.003, 2017.

Li, J., Liu, Z., Lin, B., Zhao, Y., Cao, J., Zhang, X., Zhang, J., Ling, C., Ma, P., and Wu, J.: Zooplankton fecal pellet characteristics and contribution to the deep-sea carbon export in the southern South China Sea, J. Geophys. Res.: Oceans, 127, e2022JC019412, https://doi.org/10.1029/2022jc019412, 2022.

Siegel, D. A., Buesseler, K. O., Doney, S. C., Sailley, S. F., Behrenfeld, M. J., and Boyd, P. W.: Global assessment of ocean carbon export by combining satellite observations and food-web models, Global Biogeochem. Cycles, 28, 181–196, https://doi.org/10.1002/2013GB004743, 2014.

Stamieszkin, K., Pershing, A. J., Record, N. R., Pilskaln, C. H., Dam, H. G., and Feinberg, L. R.: Size as the master trait in modeled copepod fecal pellet carbon flux, Limnol. Oceanogr., 60: 2090–2107, https://doi.org/10.1002/lno.10156, 2015.

Terrats, L., Claustre, H., Briggs, N., Poteau, A., Briat, B., Lacour, L., Ricour, F., Mangin, A., and Neukermans, G.: BioGeoChemical-Argo floats reveal Stark latitudinal gradient in the Southern Ocean deep carbon flux driven by phytoplankton community composition, Global Biogeochem. Cycles, 37, e2022GB007624, https://doi.org/10.1029/2022GB007624, 2023.

Turner, J. T.: Zooplankton fecal pellets, marine snow, phytodetritus and the ocean's biological pump, Prog. Oceanogr., 130, 205–248, https://doi.org/10.1016/j.pocean.2014.08.005, 2015.

Wang, H., Liu, Z., Li, J., Lin, B., Zhao, Y., Zhang, X., Cao, J., Zhang, J., Song, H., and Wang, W.: Sinking fate and carbon export of zooplankton fecal pellets: insights from time-series sediment trap observations in the northern South China Sea, Biogeosciences, 20, 5109–5123, https://doi.org/10.5194/bg-20-5109-2023, 2023.

Wilson, S. E., Steinberg, D. K., and Buesseler, K. O.: Changes in fecal pellet characteristics with depth as indicators of zooplankton repackaging of particles in the mesopelagic zone of the subtropical and subarctic North Pacific Ocean, Deep Sea Res. Part II, 55(14–15), 1636–1647, https://doi.org/10.1016/j.dsr2.2008.04.019, 2008.

Shatova, O., Koweek, D., Conte, M. H., and Weber, J. C.: Contribution of zooplankton fecal pellets to deep ocean particle flux in the Sargasso Sea assessed using quantitative image analysis, J. Plankton Res., 34, 905–921, https://doi.org/10.1093/plankt/fbs053, 2012.

---

## Author Response (AR3)

**Revision Note on the revised manuscript, dynamic upper-ocean processes enhance mesopelagic carbon export of zooplankton fecal pellets in the southern South China Sea, manuscript no. egusphere-2025-2864, submitted for publication in Biogeosciences.**

We sincerely thank the associate editor for the valuable comments and suggestions in improving our manuscript. This correction note is written based on the annotated (using track changes) version of the manuscript (uploaded in the system), which is a point-to-point correction note of the manuscript. In this case, we have thoroughly checked the main text, figures, and formulas to make the manuscript more proficient. The notes (in blue) explain how and where each point of comment has been addressed. The line numbers mentioned are new numbers in the annotated version of the manuscript.

**Public justification**

**#1:** It seems the legend color for summer monsoon is different from that in Figs. 3a&c.

**Reply**: Thank you for pointing this out. We have now revised the legend color for the summer and winter monsoon in Figure 3 to ensure consistency across all panels. (Line 203)

**#2:** Fig. 3: To be consistent with other figures, re-order the legends so that they start with "Summer monsoon" followed by "Winter monsoon",....

**Reply:** Thank you for this suggestion. We have now reordered the legends in Figure 3 so that "Summer monsoon" precedes "Winter monsoon", consistent with other figures. (Line 203)

**#3:** The text in Fig. 3, Fig. 7, and Fig. 8 is not sufficiently clear, particularly for that of the legends and sometime the horizontal axes as well.

**Reply:** Thank you for this comment. We have removed the text at the top of the figures and increased the font size of the legends and the x-axis labels in Figures 3, 7, and 8 to improve readability. (Lines 203, 326, 369)

In addition, we thoroughly checked the main text, figures, and formulas, and corrected a few errors to make the manuscript more proficient. (Lines 6, 11, 15, 17, 61, 157-165, 169, 235, 255, 259, 282, 304-305, 314, 331, 380, 402)